# DIFFUSION–ATTENTION CONNECTION

## ABSTRACT

Transformers, diffusion maps, and magnetic Laplacians are usually treated as separate tools; we show they are all different regimes of a single Markov geometry built from pre-softmax query–key scores. We define a QK "bidivergence" whose exponentiated and normalized forms yield attention, diffusion maps, and magnetic diffusion. And use product-of-experts and Schrödinger-bridges to connect and organize them into equilibrium, non-equilibrium steady-state, and driven dynamics.

## 1 INTRODUCTION

The evolution of neural computation can be read as a progression of increasingly expressive "artificial tissues" for representation and transformation. Early linear methods such as principal component analysis (Pearson, 1901) introduced the matrix product as a basic representational primitive. Multilayer perceptrons (Rosenblatt, 1958; Rumelhart et al., 1986) together with efficient gradient-based optimization (Werbos, 1974) established nonlinear feedforward networks as practical universal function approximators. The Transformer (Vaswani et al., 2017) established a new universal primitive: a "relational tissue" based on self-attention. By replacing fixed architectural wiring with content-based global connectivity—allowing any token to directly influence any other through learned similarities—it created a substrate that, when stacked in layers, behaves less like a handcrafted circuit and more like a continuous cortical medium. As popularized by Karpathy and others, this medium dynamically routes, mixes, and stores information, exposing useful structure from language, images, audio, or code with minimal architectural change. Most recently, Diffusion Transformers (DiTs) (Peebles & Xie, 2023) have combined diffusion-based (Ho et al., 2020; De Bortoli et al., 2021) generative dynamics with transformer-style attention. This has produced scalable generative models in which a stochastic temporal process is coupled to a globally connected spatial operator, further reinforcing the status of attention as a de facto neural tissue for general-purpose computation.

In parallel to this architectural lineage, a rich theory of nonparametric kernel methods has developed, Parzen (1962); Schölkopf & Smola (2002); Williams & Rasmussen (2005). While Transformers arrived late in this story, their attention mechanism is now often reframed in kernel-theoretic terms. One prevalent view interprets softmax attention as a form of kernel smoothing or regression, while other work, such as Teo et al. (2024), derives it from spectral objectives like kernel PCA. These interpretations, however, apply a nonlinear "kernelized" envelope to a more fundamental substrate: the pre-softmax attention logits. In this work, we take these raw query-key scores as our primary object of study. This shift in perspective allows us to recast attention in terms of divergences and Markov operators, connecting it to a broader geometric and probabilistic framework.

## 2 DIVERGENCES

To begin our discussion we wish to determine the similarity between two data samples. Samples in our dataset are characterized by a vector of numbers in some high dimensional space $v_X, w_X \in \mathbb{R}^D$. Over $N$ samples, we obtain a high-dimensional point-cloud represented by the following *dataset* matrix $R_{iX} \in \mathbb{R}^{N \times D}$, with indices $i, j \in \{1, 2, \cdots, N\}$ and $X, Y \in \{1, 2, \cdots, D\}$. The simplest similarity measure is the Gram matrix (when applied to all sample-sample, $ij$, pairs):

$$G_{ij} = \sum_X R_{iX} R_{Xj}^\top \quad .$$

This matrix is symmetric $G_{ij} = G_{ji}$, and has diagonal $G_i = G_{ii}$. In order to get Euclidean squared-distance proximity:

$$D_{ij}^2 = G_i \mathbf{1}_j + \mathbf{1}_i G_j - 2G_{ij} \tag{1}$$
$$= \underbrace{\mathbf{1}_i G_j - G_{ij}}_{i \to j} + \underbrace{G_i \mathbf{1}_j - G_{ji}}_{j \to i}$$
$$= d_{ij}^{\to} + d_{ij}^{\leftarrow} \quad . \tag{2}$$

We call the $d^{\to}$ and $d^{\leftarrow}$ partition, the two components of the Query-Key (QK) bidivergence: a pair of signed pseudo-divergences whose sum is the usual Euclidean squared-distance. Both pieces satisfy two divergence-like properties: self-zero $d_{ii}^{\to} = G_{ii} - G_{ii} = 0 = G_{ii} - G_{ii} = d_{ii}^{\leftarrow}$, and asymmetry $d_{ij}^{\leftarrow} \neq d_{ji}^{\leftarrow}$ and $d_{ij}^{\to} \neq d_{ji}^{\to}$ (however in this case $d^{\to} = (d^{\leftarrow})^{\top}$). However, each component may take negative values, but their sum is always nonnegative and equals $D^2 \geq 0$, hence the name *bidivergence*.

For a generalization of our Gram-correlation matrix we may introduce a potentially asymmetric weight matrix, we call the QK matrix denoted by $W^{\to} = (W^{\leftarrow})^{\top}$:

$$\mathcal{G}_{ij} = \sum_{X,Y} R_{iX} W_{XY} R_{Yj}^{\top} \quad . \tag{3}$$

In this case our eq. 1, becomes the Mahalanobis square-distance, this is because only the symmetric part of this QK matrix $S = \frac{1}{2}(W^{\to} + W^{\leftarrow})$ contributes. Whilst, the QK bidivergence parts are still asymmetric, and now strongly, such that $d^{\to} \neq (d^{\leftarrow})^{\top}$. A useful partition of $\mathcal{G}$, is to construct a real-symmetric and imaginary-antisymmetric terms constructing a combined complex Hermitian matrix ($V = V^{\dagger}$):

$$V = \underbrace{\frac{1}{2}(W^{\to} + W^{\leftarrow})}_{\Re V = S} + i \underbrace{\frac{1}{2}(W^{\to} - W^{\leftarrow})}_{\Im V} \quad . \tag{4}$$

This asymmetric interaction (encoded in the pieces of $d^{\leftrightarrow}$ and asymmetry in the QK matrix) yields a directed graph (with "half-edges"), this parition is especially important in sequence-modeling contexts, reflecting the fundamental observation that time has an inherent direction–*arrow-of-time*.

## 3 MARKOV OPERATORS ON DIVERGENCES

Our *divergences*, or dissimilarities, increase in value with dissimilarity. We would like to convert this into a probability distribution, and we begin by inverting this dissimilarity into a similarity with $d = \infty \to 0$, and $d = 0 \to 1$. This similarity score is achieved via: the common *Gaussian Radial-Basis-Function* (RBF), with an inverse-temperature $\beta > 0$ hyperparameter:

$$K = \exp\left(-\beta D^2\right) \quad , \tag{5}$$

and is a proper RKHS kernel. We may also form the following asymmetric operators (associated to each direction):

$$K_{ij}^{\to} = \exp\left(-\beta d_{ij}^{\to}\right) \qquad K_{ij}^{\leftarrow} = \exp\left(-\beta d_{ij}^{\leftarrow}\right) \quad . \tag{6}$$

Notably now all $K^{\to}, K^{\leftarrow}, K \geq 0$ are nonnegative, as such we may form stochastic Markov operators from these using the $\mathrm{softmax}$ (§A) and Sinkhorn (see §B) operations. In particular, $\mathrm{softmax}$ defines matrices that are either row-normalized ($\sum_j P_{ij}^+ = \mathbf{1}_i$), indicated by a superscript $+$, or column-normalized ($\sum_i P_{ij}^- = \mathbf{1}_j$), indicated by a superscript $-$. While Sinkhorn defines bistochastic matrices, and are a particular class of Markov operators $\sum_j P_{ij}^{-+} = \mathbf{1}_i$ while simultaneously $\sum_i P_{ij}^{-+} = \mathbf{1}_j$.

### 3.1 SELF-ATTENTION

As mentioned applying the softmax to our QK bidivergences ($d_{ij}^{\leftrightarrow}$) defines our two self-attention matrices:

$$A_{ij}^- = \mathrm{softmax}_i^-\left(-\beta d_{ij}^{\leftarrow}\right) \tag{7}$$
$$A_{ij}^+ = \mathrm{softmax}_j^+\left(-\beta d_{ij}^{\to}\right) \quad . \tag{8}$$

For the queries→keys $(+)$, the row-normalized form $(\sum_j A_{ij}^+ = \mathbf{1}_i)$, and the keys→queries $(-)$, column-normalized $(\sum_i A_{ij}^- = \mathbf{1}_j)$, versions. This can be shown for the most commonly used form of self-attention, $A_{ij}^+$ (with the QK bidivergence matrix, eq. 3, factorized: $W = W^{(Q)}W^K$):

$$A_{ij}^+ = \text{softmax}_j^+\left(-\beta d_{ij}^{\rightarrow}\right)$$

$$= \text{softmax}_j^+\left(-\beta\left(\mathbf{1}_i\left(RW^{(Q)}W^{(K)}R^{\top}\right)_j - R_{iY}W_{YX}^{(Q)}W_{XZ}^{(K)}R_{Zj}^{\top}\right)\right)$$

$$= \text{softmax}_j^+\left(-\beta\left(-RW^{(Q)}W^{(K)}R^{\top}\right)_{ij}\right) = \text{softmax}_j^+\left(\beta Q_{iX}K_{Xj}^{\top}\right)$$

This works because of the shift-invariance of softmax, see §A, namely $\text{softmax}_j^+(c_i + z_{ij}) = \text{softmax}_j^+(z_{ij})$, and vice-versa. In principle, we should label $A^{\rightarrow+} = A^+$, but as $+$ $(-)$ matches polarity $\rightarrow$ $(\leftarrow)$, we simplify our notation, in the case polarity does not agree we will write both superscripts. Additionally, we can apply Sinkhorn normalization, to obtain bistochastic operators:

$$A_{ij}^{-+} = \text{Sinkhorn}\left(-\beta d_{ij}^{\leftarrow}\right) \tag{9}$$

$$A_{ij}^{+-} = \text{Sinkhorn}\left(-\beta d_{ij}^{\rightarrow}\right) \quad . \tag{10}$$

In general, $A^{+-}$ and $A^{-+}$ are neither equal nor transposes of each other.

## 3.2 DIFFUSION-MAPS

Interestingly, there exists a Markov operator for the combined square-distance matrix as well:

$$P^+ = \text{softmax}^+\left(-\beta D^2\right) \tag{11}$$

$$P^- = \text{softmax}^-\left(-\beta D^2\right) \quad .$$

This operator is well-known in the literature and is known as the Diffusion-Map (DMAP) operator introduced by Coifman & Lafon (2006). This operator originates from the graph or combinatorial Laplacian:

$$L = \text{diag}(D) - K \quad , \tag{12}$$

for some weighted adjacency-matrix $K_{ij} \in \mathbb{R}^{N \times N}$, and its normalization the degree-vector $D_i = \sum_j K_{ij}$. The weighted-adjacency matrix to match our definition in eq. 11, is defined by common *Gaussian* RBF, eq. 5. Normalizing the Laplacian, by dividing by this degree-vector, we obtain the Markov-Laplacian operator:

$$L^+ = I - P^+ \quad . \tag{13}$$

Under standard assumptions, $P^+$ approximates one step of a diffusion process on a data manifold, and its eigenvectors/eigenvalues yield diffusion coordinates and an intrinsic Laplacian on the manifold. Although, the diffusion-maps operator that may be diagonalized is asymmetric due to the row-normalization, it is superficially so, as we can create a symmetric kernel that shares the same spectrum as $P^+$. Next, we can create the Sinkhorn, bistochastic, version of the diffusion-maps operator:

$$P^{+-} = \text{Sinkhorn}(-\beta D^2) = P^{-+} \quad . \tag{14 crossed?}$$

This was studied in Wang et al. (2012); Coifman & Hirn (2013). Unlike Attention, Diffusion-Maps is fundamentally symmetric relying on the symmetric and geometric distances $D^2$, however there is a generalization known as Magnetic Laplacian Eigenmaps, developed by Fanuel et al. (2017); He et al. (2023). In this theory, the complex partition, eq. 4, of the squared-distance yields two parts: a symmetric part which yields a matrix $P^+$ (like before) and an antisymmetric part which contributes to a phase matrix $U$. The magnetic-Diffusion-maps operator is thus:

$$U = \exp\left(i\Im(V)\right) \tag{14}$$

$$\tilde{P}^+ = P^+ \odot U \quad . \tag{15}$$

Once again $\tilde{P}^+$, can be converted into a Hermitian operator that may be used for diagonalization, for subsequent factorization of the entire Markov operator $\tilde{P}^+$. For brevity later, we will refer to the self-attention based theory as AMAP, analogous to DMAP, and similarly Magnetic-DMAP as MMAP.

## 4 THE SCHRÖDINGER BRIDGE CONNECTION

We now turn to another class of Markov operators arising from entropic optimal transport: discrete Schrödinger bridges (SBs), Schrödinger (1931); Di Marino & Gerolin (2020). Given a strictly positive reference kernel $K \in \mathbb{R}^{N \times N}$, e.g. the Gaussian RBF kernel equation 5, and two endpoint marginals $\mu^+, \mu^- \in \Delta^{N-1}$, the (discrete, one-step) Schrödinger bridge problem seeks a coupling $\Pi \in \mathbb{R}_{\geq 0}^{N \times N}$ that minimizes the relative entropy with respect to $K$:

$$\min_{\Pi \geq 0} \sum_{i,j} \Pi_{ij} \left( \log \tfrac{\Pi_{ij}}{K_{ij}} - 1 \right) \qquad \text{s.t.} \qquad \sum_j \Pi_{ij} = \mu_i^+, \quad \sum_i \Pi_{ij} = \mu_j^-.$$

Under mild assumptions, the minimizer exists, is unique, and has the diagonal-scaling form

$$\Pi = \operatorname{diag}(u^+) \, K \operatorname{diag}(u^-), \tag{16}$$

where $u^+, u^- \in \mathbb{R}_{>0}^N$ are the *Schrödinger potentials* (or Sinkhorn scaling factors) associated with the source $(+)$ and sink $(-)$ marginals, respectively. The Schrödinger potentials $u^\pm$ are computed as fixed points of the classical Sinkhorn (a.k.a. Schrödinger) iterations, which alternately rescale the rows and columns of the reference kernel to match the prescribed marginals, see Appendix B.4 for details. The associated *forward* Markov kernel of the bridge is obtained by normalizing along the source marginal:

$$\Pi_{ij}^+ := \frac{\Pi_{ij}}{\mu_i^+}, \qquad \sum_j \Pi_{ij}^+ = \mathbf{1}_i. \tag{17}$$

Thus $\Pi^+$ describes one step of a Markov process sending $\rho_0 = \mu^+$ to $\rho_1 = \mu^- = \rho_0 \Pi^+$.

### 4.0.1 EQUILIBRIUM, NESS, AND NONSTATIONARY BRIDGES

To characterize the dynamical regime induced by a Markov kernel, we use the notion of *probability currents*. Given a row-stochastic Markov kernel $P \in \mathbb{R}^{N \times N}$ and a probability vector $\rho \in \Delta^{N-1}$, define the antisymmetric current

$$J_{ij}(\rho) := \rho_i P_{ij} - \rho_j P_{ji}, \qquad J_{ij}(\rho) = -J_{ji}(\rho). \tag{18}$$

We say that a probability vector $\rho$ is *stationary* for a row-stochastic kernel $P$ if $\rho = \rho P$. The pair $(P, \rho)$ is at *equilibrium* (EQ) if $\rho$ is stationary and the associated probability currents vanish identically, $J_{ij}(\rho) \equiv 0$ for all $i, j$ (detailed balance). If $\rho$ is stationary but $J_{ij}(\rho) \neq 0$ for some $i, j$, then $(P, \rho)$ is called a *non-equilibrium steady state* (NESS).

### 4.1 DIFFUSION OPERATORS AS SCHRÖDINGER BRIDGES

Recall the Gaussian kernel $K_{ij} = \exp\left(-\beta D_{ij}^2\right) = K_{ji} > 0$, and the row-normalized diffusion-maps operator $Z_i^+ := \sum_j K_{ij}, P_{ij}^+ := \frac{K_{ij}}{Z_i^+}$ together define the intrinsic stationary distribution:

$$\pi_i := \frac{Z_i}{\sum_k Z_k} \quad . \tag{19}$$

It is well known (and we recall in Proposition C.2, in §C.2) that $\pi$ is stationary for $P^+$ and that detailed balance holds: $\pi = \pi P^+$, and $\pi_i P_{ij}^+ = \pi_j P_{ji}^+ \quad \forall i, j$. Thus $(P^+, \pi)$ is an EQ pair in the sense of §4.0.1.

We next embed diffusion-maps into the discrete SB framework. Consider the SB problem with $K$ as above and *equal endpoint marginals* $\mu^+ = \mu^- = \pi$. Define the coupling $\Pi_{ij} := \pi_i P_{ij}^+$. By construction, $\Pi$ has both row and column marginals equal to $\pi$. Moreover, $\Pi$ admits the SB factorization eq. 16 with:

$$u_i^+ = \frac{\pi_i}{Z_i}, \qquad u_j^- = \mathbf{1}_j, \tag{20}$$

such that $\Pi = \operatorname{diag}(u^+) \, K \operatorname{diag}(u^-)$. A formal statement and proof that $\Pi$ is in fact the unique Schrödinger bridge coupling for $(K, \mu^+, \mu^-)$, and that its forward kernel coincides with

the diffusion-maps operator $P^+$, are given in Theorem C.3 (Appendix C.2). If we replace $P^+$ by its Sinkhorn-normalized bistochastic version $P^{+-}$ from §3.2, then $\mu^+ = \mu^- = \mathbf{1}/N$ and the resulting equilibrium SB has a *uniform* stationary distribution.

More general Schrödinger bridges over the same symmetric kernel $K$ arise as Doob-transformed versions of $P^+$. Suppose $\mu^+ = \mu^- = \rho$ and $\Pi_{ij} = u_i^+ K_{ij} u_j^-$ is the corresponding SB coupling, with forward kernel:

$$\Pi_{ij}^+ = \frac{\Pi_{ij}}{\mu_i^+} = \frac{K_{ij} u_j^-}{\sum_k K_{ik} u_k^-}. \tag{21}$$

Writing $Z_i = \sum_k K_{ik}$ and $P_{ij}^+ = K_{ij}/Z_i$ as above, we obtain:

$$\Pi_{ij}^+ = P_{ij}^+ \frac{u_j^-}{(P^+ u^-)_i}. \tag{22}$$

As shown in Proposition C.4 (Appendix C.3), this is exactly a Doob $h$-transform of $P^+$ with $h = u^-$. For $h \equiv 1$ we recover the equilibrium DMAP bridge; for generic non-constant $h$, the transformed kernel is typically non-reversible and realizes a NESS SB with nonzero stationary currents $J_{ij}(\rho) \neq 0$. In softmax form, eq. 22 can be written as:

$$\Pi_{ij}^+ = \text{softmax}_j^+ \big( \log K_{ij} + \psi_j \big), \qquad \psi_j := \log u_j^-, \tag{23}$$

highlighting that $\Pi^+$ shares the same underlying geometry $K_{ij}$ as diffusion-maps, but with an additional column potential that tilts the dynamics.

When $\mu^+ \neq \mu^-$, the SB coupling still factorizes as $\Pi_{ij} = u_i^+ K_{ij} u_j^-$, and the forward kernel retains the Doob-transform structure, eq. 22. However, the induced propagation:

$$\rho_0 = \mu^+, \qquad \rho_1 = \mu^- = \rho_0 \Pi^+ \tag{24}$$

is now genuinely nonstationary (NE): there is no single invariant marginal, and the system is explicitly driven from $\mu^+$ to $\mu^-$ in one step over the same symmetric geometry $K$.

Conceptually, diffusion-maps correspond to an EQ SB over the Gaussian kernel $K$, with intrinsic marginal $\pi$ and no probability currents. General SBs over $K$ correspond to Doob-transformed (tilted) versions of this diffusion operator: EQ bridges reproduce DMAP, NESS bridges correspond to non-reversible tilts with stationary currents, and NE bridges transport between distinct marginals in one step.

### 4.2 Diffusion and attention operators

Having related DMAP and SB operators, we next connect them to the attention operators from §3.1. The starting point is the QK bidivergence decomposition $D_{ij}^2 = d_{ij}^{\rightarrow} + d_{ij}^{\leftarrow}$, which lifts the symmetric squared distance into two directed components. Applying the Gaussian RBF nonlinearity to this decomposition yields

$$K_{ij} := \exp\big(-\beta D_{ij}^2\big) = \exp(-\beta d_{ij}^{\rightarrow}) \exp(-\beta d_{ij}^{\leftarrow}) =: K_{ij}^{\rightarrow} K_{ij}^{\leftarrow} = \underbrace{z_i^+ A_{ij}^+}_{} \underbrace{A_{ij}^- z_j^-}_{}, \tag{25}$$

with directional partition functions $z_i^+ = \sum_\ell \exp\big(-\beta d_{i\ell}^{\rightarrow}\big)$, $z_j^- = \sum_\ell \exp\big(-\beta d_{\ell j}^{\leftarrow}\big)$, and using the self-attention operators from §3.1. Consequently, the row-normalized diffusion kernel $P^+$ (§4.1) satisfies

$$Z_i^+ = \sum_\ell K_{i\ell} = z_i^+ \sum_\ell z_\ell^- A_{i\ell}^+ A_{i\ell}^-,$$

and hence:

$$\boxed{P_{ij}^+ = \left( \frac{1}{\sum_\ell z_\ell^- A_{i\ell}^+ A_{i\ell}^-} \right)_i A_{ij}^+ A_{ij}^- z_j^-} . \tag{26}$$

Eq. 26 has a Schrödinger-bridge flavor: a symmetric reference kernel $K$ is "tilted" by two directional potentials $z^+, z^-$, and the resulting Markov kernel $P^+$ is obtained by combining forward

and backward conditionals multiplicatively and then normalizing. A precise derivation of eq. 26 is given in Theorem C.6 (Appendix C.4). In the special case where the column factors $z_j^-$ are (approximately) constant in $j$, the unary factor $z_j^-$ can be absorbed into the normalization, and equation 26 reduces to a pure product-of-experts (PoE) in $A^{\pm}$:

$$P_{ij}^+ \approx \frac{1}{\sum_\ell A_{i\ell}^+ A_{i\ell}^-} \, A_{ij}^+ A_{ij}^-. \tag{27}$$

An even cleaner PoE structure appears if both directional experts use the same row normalization:

$$A_{ij}^{\rightarrow +} := \mathrm{softmax}_j^+ \left( -\beta d_{ij}^{\rightarrow} \right), \qquad A_{ij}^{\leftarrow +} := \mathrm{softmax}_j^+ \left( -\beta d_{ij}^{\leftarrow} \right), \tag{28}$$

so that the backward expert now also uses the $+$ normalization (softmax over $j$). Then the diffusion kernel is simply the row-softmax of the sum of logits: $P_{ij}^+ = \mathrm{softmax}_j^+ \left( -\beta d_{ij}^{\rightarrow} - \beta d_{ij}^{\leftarrow} \right)$. Using the elementary identity (proved in Lemma C.5, Appendix C.4), we obtain the exact product-of-experts form:

$$\boxed{P_{ij}^+ = m_i^+ \, A_{ij}^{\rightarrow +} A_{ij}^{\leftarrow +}, \qquad m_i^+ := \left( \sum_\ell A_{i\ell}^{\rightarrow +} A_{i\ell}^{\leftarrow +} \right)^{-1}.} \tag{29}$$

For each row $i$, the transition probabilities $P_{i,:}^+$ are obtained by a Hadamard product of the two directional experts, followed by renormalization: $P^+ \propto (A^{\rightarrow +} \odot A^{\leftarrow +})$. Thus diffusion over the symmetric geometry $D^2$ can be viewed as a PoE combination of two directional attention maps.

### 4.2.1 MESSAGE-PASSING AND SB INTERPRETATION

The representations of equations 26 and 29 suggest a natural message-passing interpretation. $A_{ij}^{\rightarrow +}$ can be viewed as a *forward message* from $i$ to $j$, encoding which neighbors $j$ are preferred from the perspective of $d^{\rightarrow}$. $A_{ij}^{\leftarrow +}$ plays the role of a *backward message* (or future constraint) on $j$, derived from $d^{\leftarrow}$. The Markov kernel $P^+$ is the locally consistent belief obtained by combining these messages multiplicatively (PoE) and renormalizing along each row. In the Schrödinger-bridge viewpoint, the symmetric kernel $K$ is a reference dynamics (e.g. a heat kernel), the factors $z^+, z^-$ act as forward and backward potentials, and equation 26 expresses the resulting time-symmetric Markov kernel $P^+$ as the normalized product of forward and backward conditionals. Together with the constructions in §4.1 and §4.3, this shows that DMAP, attention, and SBs are different faces of the same underlying bidivergence geometry: symmetric EQ diffusion, directional NESS attention, and their PoE combinations.

### 4.3 ATTENTION OPERATORS AS SCHRÖDINGER BRIDGES

We now extend the SB perspective from the symmetric diffusion kernel (DMAP) to the intrinsically directed attention operators built from the QK bidivergences $d^{\rightarrow}, d^{\leftarrow}$ introduced in §2. In contrast to diffusion maps, which yield an EQ SB over a symmetric kernel, attention operators naturally realize *non-equilibrium steady-state* (NESS) bridges due to their built-in asymmetry.

From the forward component $d_{ij}^{\rightarrow}$ define the unnormalized transport-matrix $K_{ij}^+ := \exp(-\beta d_{ij}^{\rightarrow})$ and the corresponding (row-normalized) attention map $A^+$ which is precisely the usual (query→key) self-attention operator from §3.1. Under standard irreducibility assumptions $A^+$ admits a unique stationary distribution $\pi^+ \in \Delta^{N-1}$ such that $\pi^+ = \pi^+ A^+$. Because $K^+$ is typically asymmetric, detailed balance generically fails and the probability currents

$$J_{ij}(\pi^+) := \pi_i^+ A_{ij}^+ - \pi_j^+ A_{ji}^+ \tag{30}$$

are nonzero for many edges: $(A^+, \pi^+)$ is thus a NESS in the sense of §4.0.1. An entirely analogous construction holds for the backward component $d_{ij}^{\leftarrow}$.

We now take the forward attention logits $K^+$ as the reference kernel for a discrete SB, in direct analogy with the DMAP case but without symmetry. Given endpoint marginals $\mu^+, \mu^- \in \Delta^{N-1}$, the unique SB coupling has the factorized form; with forward SB operator:

$$\Pi_{ij}^+ := \frac{\Pi_{ij}}{\mu_i^+} = \frac{u_i^+ K_{ij}^+ u_j^-}{\mu_i^+}. \tag{31}$$

Using the row constraint $\mu_i^+ = u_i^+ \sum_j K_{ij}^+ u_j^-$, we eliminate $u_i^+$ and obtain

$$\Pi_{ij}^+ = \frac{K_{ij}^+ u_j^-}{\sum_k K_{ik}^+ u_k^-} = \text{softmax}_j^+\big(\log K_{ij}^+ + \psi_j\big), \qquad \psi_j := \log u_j^-. \tag{32}$$

Since $\log K_{ij}^+ = -\beta d_{ij}^{\rightarrow}$, the forward SB kernel is a *column-biased* attention map:

$$\Pi_{ij}^+ = \text{softmax}_j^+\big(-\beta\, d_{ij}^{\rightarrow} + \psi_j\big), \tag{33}$$

sharing the same forward geometry $d^{\rightarrow}$ as $A^+$ but with an additional key-side log-potential $\psi$. Equivalently, we can express $\Pi^+$ as a Doob transform of $A^+$. Using $A_{ij}^+ = K_{ij}^+ / \sum_\ell K_{i\ell}^+$, we get

$$\Pi_{ij}^+ = \frac{A_{ij}^+ u_j^-}{(A^+ u^-)_i}, \tag{34}$$

or in matrix form

$$\Pi^+ = \text{diag}\big((A^+ u^-)^{-1}\big)\, A^+ \text{diag}(u^-), \tag{35}$$

which is a Doob $h$-transform of $A^+$ with $h = u^-$.

A natural question is when the forward SB kernel $\Pi^+$ coincides with the un-biased attention map $A^+$. As formalized in Proposition C.7 (Appendix C.5), this happens precisely when the marginals are chosen to match one step of attention:

$$\Pi^+ = A^+ \iff \mu^- = \mu^+ A^+. \tag{36}$$

In particular, taking $\mu^+ = \mu^- = \pi^+$ gives a stationary SB with forward kernel $\Pi^+ = A^+$: forward attention can be interpreted as the forward operator of a stationary Schrödinger bridge over the asymmetric kernel $K^+$. Because detailed balance still fails, this bridge is a NESS SB, in contrast to the EQ SB realized by diffusion maps over a symmetric kernel.

### 4.4 MAGNETIC DIFFUSION-MAPS AS COMPLEX SCHRÖDINGER BRIDGES

The DMAP–SB connection in §4.1 relied crucially on a real symmetric kernel $K$. In many applications, the underlying interactions contain an intrinsically antisymmetric component. Magnetic diffusion-maps (MMAP), or magnetic Laplacian eigenmaps Fanuel et al. (2017), enrich the symmetric DMAP kernel with a complex phase field that captures such directed effects while preserving the *probability* geometry. Let's recall the symmetric Gaussian kernel, $K$, and DMAP operator $P^+$. From the complex QK interaction in eq. 4, $V = S + i\mathcal{A}$, we obtain a real symmetric part $S$ and a real antisymmetric part $\mathcal{A} = \Im(V)$, with $\mathcal{A}_{ij} = -\mathcal{A}_{ji}$. Magnetic diffusion-maps associate to $A$ a unit-modulus phase field:

$$\Theta_{ij} := \mathcal{A}_{ij}, \qquad \Theta_{ji} = -\Theta_{ij}, \tag{37}$$

$$U_{ij} := e^{i\Theta_{ij}}, \qquad U_{ji} = \overline{U_{ij}}, \tag{38}$$

and define a *magnetic* kernel and complex-valued diffusion operator

$$\tilde{K}_{ij} := K_{ij} U_{ij}, \tag{39}$$

$$\tilde{P}_{ij}^+ := \frac{\tilde{K}_{ij}}{\sum_k K_{ik}} = P_{ij}^+ U_{ij}. \tag{40}$$

Because $|U_{ij}| = 1$, the magnitudes of $\tilde{P}^+$ agree exactly with DMAP: $|\tilde{P}_{ij}^+| = P_{ij}^+, \qquad \forall i, j$. In particular, the intrinsic stationary marginal remains the same as for DMAP (i.e. $\pi$ from §4.1), and the real probability currents $J_{ij}(\pi)$ continue to vanish. At the level of *probabilities*, therefore, MMAP shares the same EQ SB structure as DMAP; the modification is purely in the complex phases. The phases $U_{ij}$ nevertheless induce a nontrivial *complex flux*. At equilibrium $\pi$, define the MMAP edge flux

$$F_{ij}^{\text{MMAP}} := \pi_i \tilde{P}_{ij}^+ = \pi_i P_{ij}^+ e^{i\Theta_{ij}}, \tag{41}$$

whose imaginary part

$$J_{ij}^{\text{MMAP}} := \Im F_{ij}^{\text{MMAP}} = \pi_i P_{ij}^+ \sin\Theta_{ij} \tag{42}$$

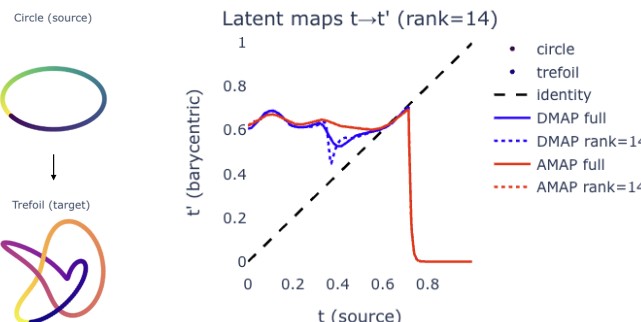

Figure 1: Circle → trefoil PoE Schrödinger bridges. Left (upper): source circle in $\mathbb{R}^3$ colored by latent coordinate $t \in [0, 1)$. Left (lower): target trefoil knot in $\mathbb{R}^3$ with the same latent parameterization. Right: barycentric latent maps $t \mapsto t'$ induced by the full Schrödinger bridge over diffusion-based (DMAP) and attention-based (AMAP) PoE kernels, together with their rank-$r$ low-rank PoE approximations (here $r = 14$). Both PoE kernels recover a smooth transport of mass along the shared latent circle.

plays the role of a *magnetic current* on the graph. This current is antisymmetric and vanishes iff the phases are trivial:

$$J_{ij}^{\text{MMAP}} = -J_{ji}^{\text{MMAP}}, \qquad \Theta_{ij} \equiv 0 \;\Rightarrow\; J_{ij}^{\text{MMAP}} \equiv 0. \tag{43}$$

Finally, the NESS probability currents of attention can be encoded as a gauge field on top of the symmetric DMAP geometry. Concretely, the antisymmetric log-flux $F_{ij}^{\text{AMAP}} := \log \frac{\pi_i^+ A_{ij}^+}{\pi_j^+ A_{ji}^+}$ defines phases $U_{ij} = e^{i\Theta_{ij}}$ which deform DMAP into a magnetic diffusion operator $\tilde{P}_{ij}^+ = P_{ij}^+ U_{ij}$ without changing the underlying probabilities. This yields a Riemann–Silberstein representation in which EQ diffusion (real part) and NESS circulation (imaginary part) coexist; detailed formulas are given in Appendix C.6.

## 5 EXPERIMENTS

**Example: circle → trefoil PoE Schrödinger bridges.** We consider two one-dimensional manifolds in $\mathbb{R}^3$, a unit circle and a trefoil knot, both parameterized by a shared latent coordinate $t \in [0, 1)$. Let $L_i(t) = \text{circle}(t_i)$ and $R_i(t) = \text{trefoil}(t_i)$ denote the corresponding embeddings (Fig. 1, left). We choose nonstationary endpoint marginals $\mu^\pm \in \Delta^{N-1}$ as Gaussian bumps in $t$ centered at distinct locations, and solve the discrete Schrödinger bridge problem $(K, \mu^+, \mu^-)$ over two different product-of-experts (PoE) kernels:

1. a diffusion-based PoE kernel $K_{\text{dmap}} = P_L \odot P_R$, where $P_L$ and $P_R$ are row-stochastic diffusion-maps kernels on $L$ and $R$, respectively;

2. an attention-based PoE kernel $K_{\text{amap}} = A_L \odot A_R$, where $A_L$ and $A_R$ are AMAP-style attention kernels derived from QK bidivergences on $L$ and $R$.

In both cases the full SB coupling $\Pi = \text{diag}(u^+) \, K \, \text{diag}(u^-)$ satisfies the prescribed marginals $\mu^\pm$ to machine precision and induces a smooth barycentric latent map $t \mapsto t'$, transporting mass along the shared circle parameter (Fig. 1, right). We visualize the barycentric latent maps $t \mapsto t'$ induced by DMAP-SB and AMAP-SB (solid curves). The dashed line is the identity baseline (no transport). Both methods transport mass from the source bump at $t \approx 0.15$ to the target bump at $t \approx 0.65$, leading to a plateau near $t' \approx 0.65$. When we truncate the PoE kernels to rank=14, AMAP-SB's latent map is almost unchanged (red dotted ≈ red solid), whereas DMAP-SB exhibits strong distortions (blue dotted vs. blue solid), indicating that AMAP kernels are much more stable under low-rank compression, see Appendix D.1 for full experimental details.

**Example: minimal DDPMs with Markov denoisers.** To test the impact of AMAP vs. DMAP geometry in a downstream generative task, we train two minimal DDPMs on a standardized 3D

Swiss roll (Fig. 2), in the spirit of recent connections between Schrödinger bridges and score-based diffusion models (De Bortoli et al., 2021). Both models share the same cosine diffusion schedule, architecture width, and optimization hyperparameters; they differ only in the Markov operator used inside the $\epsilon$-network.

In the first model (AMAP-DDPM), the denoiser consists solely of AMAP self-attention blocks, implementing time-dependent attention kernels $P_t^{\text{AMAP}}$ as in §4.2. In the second (DMAP-DDPM), we replace attention by a QK-metric DMAP kernel, yielding time-dependent diffusion operators $P_t^{\text{DMAP}}$ built from a symmetrized QK divergence, in the spirit of §4.1. Both models generate plausible samples on the Swiss-roll manifold; however, AMAP-DDPM reaches comparable sample quality (Chamfer MSE = $5.30 \times 10^{-3}$) after only $3 \times 10^4$ gradient steps, whereas the QK-DMAP-DDPM requires $1.8 \times 10^5$ steps to match it (Chamfer MSE = $4.55 \times 10^{-3}$). This supports the view that attention/AMAP Markov kernels provide a more efficient non-equilibrium geometry for learning denoising dynamics than their purely diffusion-based DMAP counterparts.

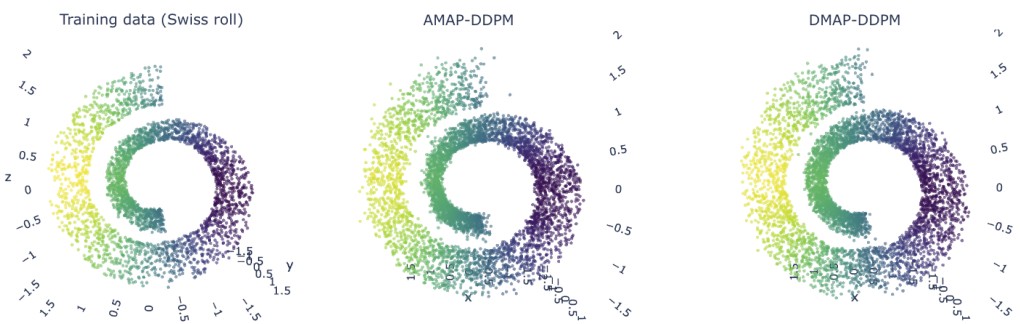

Figure 2: Minimal DDPMs with Markov denoisers on a 3D Swiss roll. Left: training data. Middle: samples from a DDPM whose $\epsilon$-network consists only of AMAP self-attention blocks (AMAP-DDPM). Right: samples from an analogous model where the self-attention is replaced by a diffusion-map kernel (DMAP-DDPM).

## 6 CONCLUSION

In this work we introduced a geometric framework that unifies attention, diffusion, and their generalizations through *QK bidivergences* and *Schrödinger bridges*. By treating pre-softmax query–key scores as directed divergences and normalizing them into Markov operators via softmax/Sinkhorn, we obtain a single Markov–geometric substrate on which self-attention (AMAP), diffusion maps (DMAP), and magnetic diffusion (MMAP) all arise as different parameterizations.

Within this framework, Schrödinger bridge theory provides a dynamical classification of Markov operators into equilibrium, non-equilibrium steady state, and nonstationary regimes, and clarifies how diffusion models implement denoising as a Markov reverse flow. The resulting product-of-experts factorization shows that diffusion kernels can be built as multiplicative combinations of directional attention maps, enabling low-rank PoE Sinkhorn layers that avoid forming dense $\mathcal{O}(N^2)$ matrices. Empirically, we find that AMAP-based PoE kernels are substantially more compressible than their DMAP counterparts and that AMAP-based denoisers learn faster in diffusion-style generative models.

Beyond the specific constructions studied here, the gauge-field and current-based viewpoint suggests a closer connection between neural architectures and nonequilibrium statistical mechanics and field theory. We view this work as a step toward such a synthesis, in which attention, diffusion, and their complex extensions are designed not as ad hoc modules but as coordinated manifestations of a single Markov–geometric substrate. A particularly promising direction is to embed our Markov-denoiser framework into large-scale latent diffusion models such as Stable Diffusion (Rombach et al., 2022), where Markov operators could serve both for geometry-aware compression into a latent space (Candanedo, 2024) and for nonlinear transport of latents (as demonstrated in this work), providing a controlled Markov–geometric alternative to conventional self-attention blocks. More

broadly, future directions include continuous-time bridge limits, scalable implementations of low-rank product-of-experts layers, and applications to nonstationary sequence models.

### SUMMARY OF CONTRIBUTIONS

1. Introduce QK bidivergence geometry for query–key scores, decomposing squared distance into directed "to/from" components.

2. Show that AMAP, DMAP, and MMAP arise as Markov operators obtained from these bidivergences via softmax/Sinkhorn normalization.

3. Use Schrödinger bridges to characterize equilibrium, non-equilibrium steady state, and nonstationary dynamical regimes of these operators.

4. Derive product-of-experts representations and low-rank PoE Sinkhorn layers for Schrödinger bridges that avoid forming full $\mathcal{O}(N^2)$ kernels.

5. Demonstrate Markov denoisers in diffusion models and empirical benefits of AMAP-based constructions in synthetic transport and generative modeling tasks.

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

# A  SOFTMAX OPERATOR

In this section, we introduce the $\mathrm{softmax}$ operator. This operator is useful to create Markov operators, that encode probability distributions, that consist of row xor column normalization while containing fully positive entries. For later use we introduce the Markov-operator shorthand:

$$z_{ij}^- := \left(\mathrm{softmax}_i^-\left(z_{ij}\right)\right)_{ij}, \qquad\qquad s_{ij}^- := \left(\mathrm{softmax}_i^-\left(s_{ij}\right)\right)_{ij}, \qquad (44)$$

$$z_{ij}^+ := \left(\mathrm{softmax}_j^+\left(z_{ij}\right)\right)_{ij}, \qquad\qquad s_{ij}^+ := \left(\mathrm{softmax}_j^+\left(s_{ij}\right)\right)_{ij}. \qquad (45)$$

Thus, the superscript "$-$" denotes the column-wise ($i$-axis) softmax and the superscript "$+$" the row-wise ($j$-axis) softmax.

**Definition A.1** (Axis-wise softmax / Boltzmann distribution). *Let $z_{ij} \in \mathbb{R}^{N \times N}$ (or be the real part of a complex matrix). The $i$-axis softmax (the Boltzmann distribution of $-z_{ij}$ with $\beta = 1$, i.e. $\mathrm{softmax}_i^-\left(-z_{ij}\right) = \mathrm{Boltzmann}_i\left(z_{ij}\right)$) is defined by elementwise exponentiation followed by normalization over the index $i$, yielding an $i$-stochastic (column-stochastic) matrix:*

$$\left(\mathrm{softmax}_i^-\left(z_{ij}\right)\right)_{ij} = \frac{e^{\odot z_{ij}}}{\mathbf{1}_k\, e^{\odot z_{kj}}} = \frac{e^{\odot z_{ij}}}{\sum_k e^{\odot z_{kj}}}. \qquad (46)$$

*Analogously, the $j$-axis softmax normalizes over the index $j$ and yields a $j$-stochastic (row-stochastic) matrix:*

$$\left(\mathrm{softmax}_j^+\left(z_{ij}\right)\right)_{ij} = \frac{e^{\odot z_{ij}}}{e^{\odot z_{ik}}\, \mathbf{1}_k} = \frac{e^{\odot z_{ij}}}{\sum_k e^{\odot z_{ik}}}. \qquad (47)$$

*In particular,*

$$\sum_i \left(\mathrm{softmax}_i^-\left(z_{ij}\right)\right)_{ij} = \sum_i z_{ij}^- = \mathbf{1}_j \quad \text{for all } j,$$

$$\sum_j \left(\mathrm{softmax}_j^+\left(z_{ij}\right)\right)_{ij} = \sum_j z_{ij}^+ = \mathbf{1}_i \quad \text{for all } i.$$

*Here $e^{\odot z_{ij}}$ denotes elementwise exponentiation of the matrix $z_{ij}$. When $z_{ij}$ is complex-valued, only its real part contributes to the normalization.*

*Theorem* A.2 (Product-of-Experts identity for softmax). Let $z_{ij}, s_{ij} \in \mathbb{R}^{N \times N}$, the $i$-axis softmax of the sum $z_{ij} + s_{ij}$ can be written as a normalized product of experts:

$$\left(\mathrm{softmax}_i^-\left(z_{ij} + s_{ij}\right)\right)_{ij} = \mu_j^-\, z_{ij}^-\, s_{ij}^-, \qquad (48)$$

where

$$\mu_j^- = \left(\sum_k z_{kj}^-\, s_{kj}^-\right)^{-1}.$$

Equivalently,

$$\left(\mathrm{softmax}_i^-\left(z_{ij} + s_{ij}\right)\right)_{ij} = \frac{z_{ij}^-\, s_{ij}^-}{\sum_k z_{kj}^-\, s_{kj}^-}. \qquad (49)$$

An analogous identity holds for the $j$-axis softmax:

$$\left(\mathrm{softmax}_j^+\left(z_{ij} + s_{ij}\right)\right)_{ij} = \mu_i^+\, z_{ij}^+\, s_{ij}^+, \qquad (50)$$

$$\mu_i^+ = \left(\sum_k z_{ik}^+\, s_{ik}^+\right)^{-1}. \qquad (51)$$

*Proof.* For each fixed column $j$ we have

$$\left(\mathrm{softmax}_i^-\left(z_{ij} + s_{ij}\right)\right)_{ij} = \frac{e^{z_{ij} + s_{ij}}}{\sum_k e^{z_{kj} + s_{kj}}} = \frac{e^{z_{ij}}\, e^{s_{ij}}}{\sum_k e^{z_{kj}}\, e^{s_{kj}}}. \qquad (52)$$

By definition of the column-wise softmax,

$$z_{ij}^- \, s_{ij}^- = \frac{e^{z_{ij}}}{\sum_k e^{z_{kj}}} \frac{e^{s_{ij}}}{\sum_k e^{s_{kj}}}.$$

$$\sum_i z_{ij}^- \, s_{ij}^- = \frac{\sum_i e^{z_{ij}} \, e^{s_{ij}}}{\left(\sum_k e^{z_{kj}}\right)\left(\sum_k e^{s_{kj}}\right)}.$$

Therefore

$$\frac{z_{ij}^- \, s_{ij}^-}{\sum_k z_{kj}^- \, s_{kj}^-} = \frac{e^{z_{ij}} \, e^{s_{ij}}}{\sum_k e^{z_{kj}} \, e^{s_{kj}}} = \left(\mathrm{softmax}_i^-\left(z_{ij} + s_{ij}\right)\right)_{ij}, \tag{53}$$

which is the claimed Product-of-Experts identity for the $i$-axis softmax. The statement for $\mathrm{softmax}_j^+ \left(\cdot\right)$ follows by symmetry, normalizing over $j$ instead of $i$, and yields the identity in terms of $z_{ij}^+$ and $s_{ij}^+$. $\qquad\square$

**Corollary A.3** (Shift invariance)**.** *Consider the special case $s_{ij} = \mathbf{1}_i \, c_j$, which is constant in $i$ for each fixed $j$ (and analogously $s_{ij} = u_i \, \mathbf{1}_j$ for shifts along the $j$-axis). Then*

$$s_{ij}^- = \left(\mathrm{softmax}_i^-\left(s_{ij}\right)\right)_{ij} = \frac{e^{c_j}}{\sum_k e^{c_j}} = \frac{1}{N}, \tag{54}$$

*so $s_{ij}^-$ is uniform in $i$. Applying the Product-of-Experts identity,*

$$\left(\mathrm{softmax}_i^-\left(z_{ij} + s_{ij}\right)\right)_{ij} = \frac{z_{ij}^- \, (1/N)}{\sum_k z_{kj}^- \, (1/N)} = z_{ij}^- = \left(\mathrm{softmax}_i^-\left(z_{ij}\right)\right)_{ij}. \tag{55}$$

*Hence softmax has a simple shift-invariance along each axis:*

$$\mathrm{softmax}_i^-\left(z_{ij} + \mathbf{1}_i \, c_j\right) = \mathrm{softmax}_i^-\left(z_{ij}\right), \tag{56}$$

$$\mathrm{softmax}_j^+\left(z_{ij} + u_i \, \mathbf{1}_j\right) = \mathrm{softmax}_j^+\left(z_{ij}\right), \tag{57}$$

*where $\mathbf{1}_i$ (resp. $\mathbf{1}_j$) denotes the all-ones vector in the $i$- (resp. $j$-) direction.*

# B    SINKHORN OPERATOR AND BISTOCHASTIC MATRICES

This appendix summarizes the Sinkhorn operator and its basic properties, which play a key role in our analysis.

**Definition B.1** (Bistochastic matrix). *A matrix $Z_{ij} \in \mathbb{R}^{N \times N}$ is* bistochastic *(or* doubly stochastic*) if $Z_{ij} \geq 0$ and*

$$\sum_i Z_{ij} = 1 \quad \text{for all } j, \qquad \sum_j Z_{ij} \qquad\qquad = 1 \quad \text{for all } i. \tag{58}$$

*Equivalently, $Z\,\mathbf{1} = \mathbf{1}$ and $\mathbf{1}^\top Z = \mathbf{1}^\top$, where $\mathbf{1}$ is the all-ones vector. The set of all $N \times N$ bistochastic matrices is the* Birkhoff polytope*; its extreme points are the permutation matrices (Birkhoff–von Neumann theorem).*

*Theorem* B.2 (Sinkhorn's theorem). Let $K_{ij} \in \mathbb{R}^{N \times N}$ have strictly positive entries. Then there exist unique diagonal matrices $D_r, D_c$ with positive diagonals (up to a common scalar factor) such that $Z_{ij} = (D_r K D_c)_{ij}$ is bistochastic. For non-negative $K$, similar results hold under mild conditions (e.g., total support).

*Proof.* Proof found in standard references: Sinkhorn (1964); Sinkhorn & Knopp (1967). □

## B.1    THE SINKHORN OPERATOR

**Definition B.3** (Sinkhorn operator). *Given a matrix of log-scores $z_{ij} \in \mathbb{R}^{N \times N}$, define the positive weight matrix $K_{ij} := \exp(z_{ij})$. The* Sinkhorn operator *returns the unique bistochastic matrix obtained from $K$ via Sinkhorn scaling:*

$$\text{Sinkhorn}(z_{ij}) := Z_{ij}, \qquad Z_{ij} = \exp\left(z_{ij} + u_i + v_j\right), \tag{59}$$

*where the vectors $(u_i)_i$, $(v_j)_j$ are the scaling potentials (unique up to an additive constant).*

## B.2    SINKHORN ITERATIONS

In practice, the Sinkhorn operator is computed by alternating row and column normalizations, known as *Sinkhorn iterations*. Starting from $Z_{ij}^{(0)} := \exp(z_{ij})$, we define for $t = 0, 1, 2, \dots$:

$$Z_{ij}^{(2t+1)} := \frac{Z_{ij}^{(2t)}}{\sum_k Z_{kj}^{(2t)}} \qquad\qquad \text{(column normalization)}, \tag{60}$$

$$Z_{ij}^{(2t+2)} := \frac{Z_{ij}^{(2t+1)}}{\sum_k Z_{ik}^{(2t+1)}} \qquad\qquad \text{(row normalization)}. \tag{61}$$

Under the conditions of Sinkhorn's theorem, $Z^{(t)} \to \text{Sinkhorn}(z_{ij})$. We denote a finite-step approximation by $\text{Sinkhorn}_T(z_{ij}) := Z_{ij}^{(T)}$.

**Implementation remarks.**

- Each step can be implemented stably in the log-domain using log-sum-exp operations; column normalization subtracts $\log \sum_k \exp(z_{kj})$ from column $j$, row normalization subtracts $\log \sum_k \exp(z_{ik})$ from row $i$.
- $\text{Sinkhorn}_T$ with small $T$ serves as a differentiable relaxation of permutation matrices in machine-learning applications.

## B.3    KEY PROPERTIES

**Lemma B.4** (Gauge invariance). *For any vectors $(u_i)_i$, $(v_j)_j$,*

$$\text{Sinkhorn}(z_{ij} + u_i + v_j) = \text{Sinkhorn}(z_{ij}). \tag{62}$$

*Proof.* If $K = \exp(z_{ij})$, then $\exp(z_{ij}+u_i+v_j) = D_u K D_v$ with diagonal matrices $D_u, D_v$. Scaling $K$ by $D_u$ and $D_v$ can be absorbed into the Sinkhorn factors $D_r, D_c$, leaving the final bistochastic matrix unchanged. □

**Lemma B.5** (Closure under multiplication). *If $A$ and $B$ are bistochastic, then $C = AB$ is also bistochastic.*

*Proof.* Non-negativity is preserved. Moreover, $C\mathbf{1} = A(B\mathbf{1}) = A\mathbf{1} = \mathbf{1}$ and $\mathbf{1}^\mathsf{T}C = (\mathbf{1}^\mathsf{T}A)B = \mathbf{1}^\mathsf{T}B = \mathbf{1}^\mathsf{T}$. □

**Corollary B.6.** *If $Z = \mathrm{Sinkhorn}(z_{ij})$ and $W = \mathrm{Sinkhorn}(s_{ij})$, then $ZW$ is bistochastic. Hence the image of* $\mathrm{Sinkhorn}$ *is closed under matrix multiplication.*

### B.4 GENERALIZATION TO SCHRÖDINGER BRIDGES

The Sinkhorn iterations are a special case of the *Schrödinger iterations* used to compute the discrete Schrödinger bridge. Given a positive reference kernel $K_{ij} > 0$ and target marginals $\mu^+, \mu^- \in \Delta^{N-1}$, the Schrödinger bridge coupling has the factored form $\Pi_{ij} = u_i^+ K_{ij} u_j^-$, where the potentials $u^+, u^- > 0$ satisfy the marginal constraints:

$$u_i^+ \sum_j K_{ij} u_j^- = \mu_i^+, \qquad u_j^- \sum_i K_{ij} u_i^+ = \mu_j^-. \tag{63}$$

These are solved by the alternating updates

$$u_{(k+1)}^+ = \frac{\mu^+}{K u_{(k)}^-}, \qquad u_{(k+1)}^- = \frac{\mu^-}{K^\mathsf{T} u_{(k+1)}^+}, \tag{64}$$

where division is componentwise. Sinkhorn scaling corresponds to the case $\mu^+ = \mu^- = \mathbf{1}/N$ (uniform marginals). Under standard conditions, the iterations converge to the unique Schrödinger bridge coupling.

## C   SCHRÖDINGER BRIDGES, DIFFUSION, AND ATTENTION: TECHNICAL RESULTS

### C.1   STATIONARITY AND EQ/NESS CLASSIFICATION

**Proposition C.1** (Stationarity of equal-marginal SBs). *Let $K > 0$ be a reference kernel, $\mu^+, \mu^- \in \Delta^{N-1}$, and $\Pi = \mathrm{diag}(u^+) K \mathrm{diag}(u^-)$ the unique Schrödinger bridge coupling with forward kernel $\Pi_{ij}^+ := \Pi_{ij}/\mu_i^+$. If $\mu^+ = \mu^- =: \rho$, then $\rho$ is stationary for $\Pi^+$:*

$$\rho = \rho\Pi^+. \tag{65}$$

*Proof.* By definition of $\Pi^+$ and the marginal constraints,

$$(\rho\Pi^+)_j = \sum_i \rho_i \Pi_{ij}^+ = \sum_i \mu_i^+ \frac{\Pi_{ij}}{\mu_i^+} = \sum_i \Pi_{ij} = \mu_j^- = \rho_j.$$

Thus $\rho$ is invariant for $\Pi^+$. □

### C.2   DIFFUSION MAPS AS EQUILIBRIUM SCHRÖDINGER BRIDGES

**Proposition C.2** (Reversibility of diffusion maps). *Let $K \in \mathbb{R}^{N \times N}$ be a strictly positive symmetric kernel, $K_{ij} = K_{ji} > 0$, and define*

$$Z_i := \sum_j K_{ij}, \qquad P_{ij}^+ := \frac{K_{ij}}{Z_i},$$

$$\pi_i := \frac{Z_i}{\sum_k Z_k}.$$

*Then $\pi$ is stationary for $P^+$ and detailed balance holds:*

$$\pi = \pi P^+, \qquad \pi_i P_{ij}^+ = \pi_j P_{ji}^+ \quad \forall i, j. \tag{66}$$

*Proof.* We have

$$\pi_i P_{ij}^+ = \frac{Z_i}{\sum_k Z_k} \cdot \frac{K_{ij}}{Z_i} = \frac{K_{ij}}{\sum_k Z_k},$$

$$\pi_j P_{ji}^+ = \frac{Z_j}{\sum_k Z_k} \cdot \frac{K_{ji}}{Z_j} = \frac{K_{ji}}{\sum_k Z_k} = \frac{K_{ij}}{\sum_k Z_k},$$

using symmetry $K_{ij} = K_{ji}$. Hence $\pi_i P_{ij}^+ = \pi_j P_{ji}^+$ for all $i, j$, and summing over $i$ gives

$$(\pi P^+)_j = \sum_i \pi_i P_{ij}^+ = \sum_i \pi_j P_{ji}^+ = \pi_j.$$

Thus $\pi$ is stationary and detailed balance holds. □

*Theorem* C.3 (Diffusion maps as an equilibrium Schrödinger bridge). Let $K$ be as in Proposition C.2 and $P^+, \pi$ defined there. Consider the coupling

$$\Pi_{ij} := \pi_i P_{ij}^+. \tag{67}$$

Then:

1. $\Pi$ has marginals $\mu^+ = \mu^- = \pi$:

$$\sum_j \Pi_{ij} = \pi_i, \qquad \sum_i \Pi_{ij} = \pi_j.$$

2. $\Pi$ admits the SB factorization equation 16 with

$$u_i^+ = \frac{\pi_i}{Z_i}, \qquad u_j^- = 1,$$

so that $\Pi = \mathrm{diag}(u^+) \, K \, \mathrm{diag}(u^-)$.

3. $\Pi$ is the unique Schrödinger bridge coupling for the triplet $(K, \mu^+, \mu^-)$, and its forward kernel $\Pi_{ij}^+ := \Pi_{ij}/\mu_i^+$ coincides with the diffusion-maps operator:

$$\Pi_{ij}^+ = P_{ij}^+.$$

*Proof.* For (1), using the definition of $\Pi$,

$$\sum_j \Pi_{ij} = \sum_j \pi_i P_{ij}^+ = \pi_i \sum_j P_{ij}^+ = \pi_i,$$

$$\sum_i \Pi_{ij} = \sum_i \pi_i P_{ij}^+ = (\pi P^+)_j = \pi_j,$$

where the last equality uses Proposition C.2. Thus $\mu^+ = \mu^- = \pi$.

For (2), note that

$$u_i^+ K_{ij} u_j^- = \frac{\pi_i}{Z_i} K_{ij} = \pi_i \frac{K_{ij}}{Z_i} = \pi_i P_{ij}^+ = \Pi_{ij},$$

so $\Pi = \mathrm{diag}(u^+) K \mathrm{diag}(u^-)$ as claimed.

For (3), the discrete SB problem with strictly positive kernel $K$ and marginals $(\mu^+, \mu^-)$ is strictly convex in $\Pi$, and its minimizer is characterized exactly by the diagonal scaling form $\Pi = \mathrm{diag}(u^+) K \mathrm{diag}(u^-)$ together with the marginal constraints Coifman & Lafon (2006); Léonard (2014). Since the pair $(u^+, u^-)$ above satisfies these constraints, the corresponding $\Pi$ is the unique SB coupling. The forward kernel is

$$\Pi_{ij}^+ = \frac{\Pi_{ij}}{\mu_i^+} = \frac{\pi_i P_{ij}^+}{\pi_i} = P_{ij}^+.$$

$\square$

### C.3 DOOB TRANSFORMS AND SB FORWARD KERNELS

**Proposition C.4** (SB forward kernels as Doob $h$-transforms). *Let $K > 0$ be any reference kernel, define $Z_i := \sum_j K_{ij}$ and $P_{ij}^+ := K_{ij}/Z_i$. Let $\Pi = \mathrm{diag}(u^+) K \mathrm{diag}(u^-)$ be a Schrödinger bridge coupling with marginals $\mu^+, \mu^- \in \Delta^{N-1}$, and forward kernel $\Pi_{ij}^+ := \Pi_{ij}/\mu_i^+$. Then*

$$\Pi_{ij}^+ = P_{ij}^+ \frac{u_j^-}{(P^+ u^-)_i}, \tag{68}$$

*where $(P^+ u^-)_i = \sum_k P_{ik}^+ u_k^-$.*

*Proof.* By definition,

$$\Pi_{ij}^+ = \frac{\Pi_{ij}}{\mu_i^+} = \frac{u_i^+ K_{ij} u_j^-}{\mu_i^+}.$$

The row-marginal constraint reads

$$\mu_i^+ = \sum_j \Pi_{ij} = u_i^+ \sum_j K_{ij} u_j^-,$$

so

$$\Pi_{ij}^+ = \frac{K_{ij} u_j^-}{\sum_k K_{ik} u_k^-}.$$

Using $K_{ij} = Z_i P_{ij}^+$ and $\sum_k K_{ik} u_k^- = Z_i (P^+ u^-)_i$, we obtain

$$\Pi_{ij}^+ = \frac{Z_i P_{ij}^+ u_j^-}{Z_i (P^+ u^-)_i} = P_{ij}^+ \frac{u_j^-}{(P^+ u^-)_i}.$$

$\square$

The expression in Proposition C.4 is a Doob $h$-transform of $P^+$ with $h = u^-$: the dynamics are reweighted by the harmonic function $h$ and renormalized rowwise.

### C.4 PRODUCT-OF-EXPERTS FACTORIZATION FOR DIFFUSION KERNELS

**Lemma C.5** (Row-softmax product-of-experts identity). *Let $(x_{ij})$ and $(y_{ij})$ be real matrices, and for each fixed $i$ define*

$$A_{ij} := \text{softmax}_j^+(x_{ij}) = \frac{e^{x_{ij}}}{\sum_\ell e^{x_{i\ell}}},$$

$$B_{ij} := \text{softmax}_j^+(y_{ij}) = \frac{e^{y_{ij}}}{\sum_\ell e^{y_{i\ell}}}.$$

*Then for all $i, j$,*

$$\text{softmax}_j^+(x_{ij} + y_{ij}) = \frac{A_{ij} B_{ij}}{\sum_\ell A_{i\ell} B_{i\ell}}. \tag{69}$$

*Proof.* For fixed $i$,

$$\text{softmax}_j^+(x_{ij} + y_{ij}) = \frac{e^{x_{ij} + y_{ij}}}{\sum_\ell e^{x_{i\ell} + y_{i\ell}}}$$

$$= \frac{e^{x_{ij}} e^{y_{ij}}}{\sum_\ell e^{x_{i\ell}} e^{y_{i\ell}}}.$$

Using the definitions of $A_{ij}, B_{ij}$,

$$A_{ij} B_{ij} = \frac{e^{x_{ij}}}{\sum_\ell e^{x_{i\ell}}} \cdot \frac{e^{y_{ij}}}{\sum_\ell e^{y_{i\ell}}} = \frac{e^{x_{ij}} e^{y_{ij}}}{(\sum_\ell e^{x_{i\ell}})(\sum_\ell e^{y_{i\ell}})},$$

and similarly

$$\sum_\ell A_{i\ell} B_{i\ell} = \sum_\ell \frac{e^{x_{i\ell}} e^{y_{i\ell}}}{(\sum_m e^{x_{im}})(\sum_n e^{y_{in}})} = \frac{\sum_\ell e^{x_{i\ell}} e^{y_{i\ell}}}{(\sum_m e^{x_{im}})(\sum_n e^{y_{in}})}.$$

Therefore

$$\frac{A_{ij} B_{ij}}{\sum_\ell A_{i\ell} B_{i\ell}} = \frac{e^{x_{ij}} e^{y_{ij}}}{\sum_\ell e^{x_{i\ell}} e^{y_{i\ell}}} = \text{softmax}_j^+(x_{ij} + y_{ij}),$$

as claimed. □

*Theorem* C.6 (Diffusion kernel as a product of directional experts). Let $D_{ij}^2 = d_{ij}^\rightarrow + d_{ij}^\leftarrow$ be the QK bidivergence decomposition of the squared distance, and define

$$K_{ij} := \exp(-\beta D_{ij}^2),$$

$$K_{ij}^\rightarrow := \exp(-\beta d_{ij}^\rightarrow), \qquad K_{ij}^\leftarrow := \exp(-\beta d_{ij}^\leftarrow),$$

so that $K_{ij} = K_{ij}^\rightarrow K_{ij}^\leftarrow$. Let

$$z_i^+ := \sum_\ell e^{-\beta d_{i\ell}^\rightarrow},$$

$$z_j^- := \sum_\ell e^{-\beta d_{\ell j}^\leftarrow},$$

$$Z_i^- := \sum_\ell e^{-\beta D_{i\ell}^2},$$

and define directional attention operators

$$A_{ij}^+ := \text{softmax}_j^+(-\beta d_{ij}^\rightarrow),$$

$$A_{ij}^- := \text{softmax}_i^-(-\beta d_{ij}^\leftarrow).$$

Then the row-normalized diffusion kernel

$$P_{ij}^+ := \frac{K_{ij}}{\sum_\ell K_{i\ell}}$$

admits the factorization

$$P_{ij}^+ = \left( \frac{1}{\sum_\ell z_\ell^- \, A_{i\ell}^+ A_{i\ell}^-} \right)_i A_{ij}^+ A_{ij}^- \, z_j^- . \tag{70}$$

Moreover, if we instead define

$$A_{ij}^{\rightarrow+} := \mathrm{softmax}_j^+(-\beta d_{ij}^{\rightarrow}),$$
$$A_{ij}^{\leftarrow+} := \mathrm{softmax}_j^+(-\beta d_{ij}^{\leftarrow}),$$

then

$$P_{ij}^+ = m_i^+ \, A_{ij}^{\rightarrow+} A_{ij}^{\leftarrow+}, \qquad m_i^+ := \left( \sum_\ell A_{i\ell}^{\rightarrow+} A_{i\ell}^{\leftarrow+} \right)^{-1} . \tag{71}$$

*Proof.* We first derive the factorization involving $A^+$ and $A^-$. By definition,

$$K_{ij}^{\rightarrow} = e^{-\beta d_{ij}^{\rightarrow}} = z_i^+ A_{ij}^+,$$
$$K_{ij}^{\leftarrow} = e^{-\beta d_{ij}^{\leftarrow}},$$

and the column partition function $z_j^-$ can be written as

$$z_j^- = \sum_\ell e^{-\beta d_{\ell j}^{\leftarrow}} = \sum_\ell K_{\ell j}^{\leftarrow} .$$

Using the definition of the column-normalized operator $A_{ij}^-$, we obtain

$$K_{ij}^{\leftarrow} = e^{-\beta d_{ij}^{\leftarrow}} = z_j^- A_{ij}^-,$$

since

$$A_{ij}^- = \frac{e^{-\beta d_{ij}^{\leftarrow}}}{\sum_\ell e^{-\beta d_{\ell j}^{\leftarrow}}} = \frac{K_{ij}^{\leftarrow}}{z_j^-} .$$

Thus

$$K_{ij} = K_{ij}^{\rightarrow} K_{ij}^{\leftarrow} = z_i^+ z_j^- \, A_{ij}^+ A_{ij}^- .$$

Summing over $j$,

$$\sum_\ell K_{i\ell} = z_i^+ \sum_\ell z_\ell^- \, A_{i\ell}^+ A_{i\ell}^- .$$

Therefore

$$P_{ij}^+ = \frac{K_{ij}}{\sum_\ell K_{i\ell}} = \frac{z_i^+ z_j^- \, A_{ij}^+ A_{ij}^-}{z_i^+ \sum_\ell z_\ell^- \, A_{i\ell}^+ A_{i\ell}^-}$$
$$= \left( \frac{1}{\sum_\ell z_\ell^- \, A_{i\ell}^+ A_{i\ell}^-} \right)_i A_{ij}^+ A_{ij}^- \, z_j^- ,$$

as claimed.

For the pure product-of-experts form, we directly use $d_{ij}^{\rightarrow} + d_{ij}^{\leftarrow} = D_{ij}^2$ and Lemma C.5. Define

$$A_{ij}^{\rightarrow+} := \mathrm{softmax}_j^+(-\beta d_{ij}^{\rightarrow}),$$
$$A_{ij}^{\leftarrow+} := \mathrm{softmax}_j^+(-\beta d_{ij}^{\leftarrow}).$$

Then

$$P_{ij}^+ = \mathrm{softmax}_j^+(-\beta D_{ij}^2) = \mathrm{softmax}_j^+(-\beta d_{ij}^{\rightarrow} - \beta d_{ij}^{\leftarrow})$$
$$= \frac{A_{ij}^{\rightarrow+} A_{ij}^{\leftarrow+}}{\sum_\ell A_{i\ell}^{\rightarrow+} A_{i\ell}^{\leftarrow+}} = m_i^+ \, A_{ij}^{\rightarrow+} A_{ij}^{\leftarrow+},$$

where

$$m_i^+ := \left( \sum_\ell A_{i\ell}^{\rightarrow+} A_{i\ell}^{\leftarrow+} \right)^{-1}$$

ensures row normalization. $\qquad\square$

### C.5 ATTENTION AS A SCHRÖDINGER BRIDGE OVER ASYMMETRIC KERNELS

**Proposition C.7** (Attention as an SB forward kernel). *Let $d_{ij}^{\rightarrow}$ be a forward QK bidivergence and*

$$K_{ij}^+ := \exp(-\beta d_{ij}^{\rightarrow}),$$

$$A_{ij}^+ := \mathrm{softmax}_j^+(-\beta d_{ij}^{\rightarrow}) = \frac{K_{ij}^+}{\sum_\ell K_{i\ell}^+}.$$

*Fix a source marginal $\mu^+ \in \Delta^{N-1}$ and set*

$$\mu^- := \mu^+ A^+.$$

*Consider the SB problem with reference kernel $K^+$ and endpoint marginals $(\mu^+, \mu^-)$. Then the unique SB coupling has forward kernel*

$$\Pi^+ = A^+. \tag{72}$$

*Conversely, if for some marginals $(\mu^+, \mu^-)$ the SB forward kernel satisfies $\Pi^+ = A^+$, then necessarily*

$$\mu^- = \mu^+ A^+. \tag{73}$$

*Equivalently, for fixed reference $K^+$,*

$$\Pi^+ = A^+ \quad \Longleftrightarrow \quad \mu^- = \mu^+ A^+. \tag{74}$$

*Proof.* (*If* part.) Define

$$u_j^- := 1, \qquad u_i^+ := \frac{\mu_i^+}{\sum_\ell K_{i\ell}^+}.$$

Then the coupling

$$\Pi_{ij} := u_i^+ K_{ij}^+ u_j^- = \frac{\mu_i^+}{\sum_\ell K_{i\ell}^+} K_{ij}^+ = \mu_i^+ A_{ij}^+$$

is of the SB form equation 16. Its row marginals are

$$\sum_j \Pi_{ij} = \sum_j \mu_i^+ A_{ij}^+ = \mu_i^+,$$

and the column marginals are

$$\sum_i \Pi_{ij} = \sum_i \mu_i^+ A_{ij}^+ = (\mu^+ A^+)_j = \mu_j^-.$$

Thus $\Pi$ is feasible for the SB problem with $(K^+, \mu^+, \mu^-)$, and by uniqueness of the discrete Schrödinger bridge for strictly positive $K^+$ it must be the optimal coupling. The forward kernel is

$$\Pi_{ij}^+ = \frac{\Pi_{ij}}{\mu_i^+} = \frac{\mu_i^+ A_{ij}^+}{\mu_i^+} = A_{ij}^+.$$

(*Only-if* part.) Suppose the SB forward kernel satisfies $\Pi^+ = A^+$. Then the coupling is $\Pi_{ij} = \mu_i^+ \Pi_{ij}^+ = \mu_i^+ A_{ij}^+$. The column marginal constraint gives

$$\mu_j^- = \sum_i \Pi_{ij} = \sum_i \mu_i^+ A_{ij}^+ = (\mu^+ A^+)_j,$$

so $\mu^- = \mu^+ A^+$. □

### C.6 ATTENTION–MAGNETIC DIFFUSION DUALITY: A RIEMANN–SILBERSTEIN VIEW

We finally connect the intrinsically NESS self-attention dynamics from §4.3 to the magnetic diffusion picture above. The key idea is that the NESS probability currents of attention can be encoded as a gauge field on top of the symmetric DMAP geometry, yielding a Riemann–Silberstein (RS) representation in which EQ diffusion (real part) and NESS circulation (imaginary part) coexist.

Recall that the forward attention kernel $A^+$ defines an intrinsic NESS with stationary marginal $\pi^+ \in \Delta^{N-1}$ and current

$$J_{ij}^{\mathrm{AMAP}} := \pi_i^+ A_{ij}^+ - \pi_j^+ A_{ji}^+. \tag{75}$$

A convenient antisymmetric log-flux is

$$F_{ij}^{\mathrm{AMAP}} := \log \frac{\pi_i^+ A_{ij}^+}{\pi_j^+ A_{ji}^+}, \qquad F_{ij}^{\mathrm{AMAP}} = -F_{ji}^{\mathrm{AMAP}}, \tag{76}$$

which vanishes exactly at equilibrium.

To transport this NESS structure onto the symmetric DMAP geometry, we use $F^{\mathrm{AMAP}}$ to define a phase field for MMAP. Choose a scale parameter $\gamma \in \mathbb{R}$ and set

$$\Theta_{ij} := \gamma F_{ij}^{\mathrm{AMAP}}, \qquad \Theta_{ji} = -\Theta_{ij}, \tag{77}$$

$$U_{ij} := e^{i\Theta_{ij}}, \qquad U_{ji} = \overline{U_{ij}}. \tag{78}$$

Using these phases together with the symmetric kernel $K$ from §3.2, we form the magnetic operator

$$\tilde{P}_{ij}^+ = P_{ij}^+ U_{ij} \tag{79}$$

as in equation 40. The resulting MMAP flux at the DMAP equilibrium $\pi$ is

$$F_{ij}^{\mathrm{MMAP}} := \pi_i \tilde{P}_{ij}^+ = \pi_i P_{ij}^+ e^{i\Theta_{ij}}, \tag{80}$$

with imaginary part

$$J_{ij}^{\mathrm{MMAP}} := \Im F_{ij}^{\mathrm{MMAP}} = \pi_i P_{ij}^+ \sin(\gamma F_{ij}^{\mathrm{AMAP}}). \tag{81}$$

For small $\gamma$,

$$\sin(\gamma F_{ij}^{\mathrm{AMAP}}) \approx \gamma F_{ij}^{\mathrm{AMAP}}, \tag{82}$$

so that

$$J_{ij}^{\mathrm{MMAP}} \approx \gamma \, \pi_i P_{ij}^+ \, F_{ij}^{\mathrm{AMAP}}. \tag{83}$$

Up to the weights $\pi_i P_{ij}^+$ and scale $\gamma$, the sign and cycle structure of $J^{\mathrm{MMAP}}$ mirrors that of the attention NESS log-flux $F_{ij}^{\mathrm{AMAP}}$ and thus of the probability current $J^{\mathrm{AMAP}}$.

## C.7 Gauge Invariance

*Theorem* C.8 (Abelian Gauge Equivariance of the Hadamard Product). Let $G = \mathrm{U}(1)^N$ be the *abelian gauge group* acting on an $N$-node graph. Let $\phi = (\phi_1, \ldots, \phi_N) \in [0, 2\pi)^N$ be a gauge choice, and define the diagonal *gauge transformation matrix*

$$R(\phi) = \mathrm{diag}\left(e^{i\phi_1}, \ldots, e^{i\phi_N}\right).$$

An *edge matrix* $E \in \mathbb{C}^{N \times N}$ is said to have *gauge weight* $\alpha \in \mathbb{R}$ if it transforms under $G$ as

$$E \longmapsto E' = R(\phi)^\alpha \, E \, R(\phi)^{-\alpha},$$

where the matrix product is interpreted entrywise:

$$E'_{ij} = e^{i\alpha(\phi_i - \phi_j)} \, E_{ij}.$$

Then the following properties hold:

1. **Closure of the Hadamard product under gauge addition**. If $B$ has weight $\alpha$ and $C$ has weight $\beta$, then their Hadamard product $(B \odot C)_{ij} := B_{ij} C_{ij}$ has weight $\alpha + \beta$:

$$(B \odot C)'_{ij} = e^{i(\alpha+\beta)(\phi_i - \phi_j)} \, (B \odot C)_{ij}.$$

2. **Gauge invariance of the adjacency structure**. Let $\Omega \in \mathbb{R}^{N \times N}$ be a real symmetric adjacency matrix (weight 0). Then the *magnitude–phase factorization*

$$\tilde{\Omega} := \Omega \odot U,$$

where $U_{ij} = e^{i\theta_{ij}}$ is a unitary edge connection of weight 1, yields a gauge–covariant operator of weight 1.

3. **Spectrum under similarity gauge action**. For any weight–0 operator $\Omega$, the similarity transformation

$$\Omega \mapsto R(\phi) \, \Omega \, R(\phi)^{-1}$$

preserves the spectrum of $\Omega$. Consequently, all spectral constructions (e.g., Laplacian eigenvalues, diffusion coordinates) are *gauge–invariant*.

*Proof.*    1. *Weight addition:*

$$(B \odot C)'_{ij} = \left(e^{i\alpha(\phi_i - \phi_j)} B_{ij}\right) \left(e^{i\beta(\phi_i - \phi_j)} C_{ij}\right) = e^{i(\alpha+\beta)(\phi_i - \phi_j)} (B \odot C)_{ij}.$$

2. *Covariance of $\tilde{\Omega}$:* $\Omega$ has weight 0, $U$ has weight 1; by (i), $\tilde{\Omega} = \Omega \odot U$ has weight 1.

3. *Spectral invariance:* The map $\Omega \mapsto R\Omega R^{-1}$ is a similarity, hence preserves eigenvalues. If $\Omega v = \lambda v$, then $(R\Omega R^{-1})(Rv) = \lambda(Rv)$.

$\square$

**Corollary C.9** (Application to Magnetic Laplacians). *Given a symmetric adjacency $\Omega$ and a unitary connection $U$ of weight* 1*, the* magnetic Laplacian

$$L_U := D - \tilde{\Omega}, \qquad \tilde{\Omega} = \Omega \odot U,$$

*where $D$ is the degree matrix, transforms under a gauge change $\phi$ as*

$$L_U \longmapsto R(\phi) \, L_U \, R(\phi)^{-1},$$

*and therefore its spectrum is* gauge–invariant.

Now that our Abelian bridge is complete, we consider the feasibility of a non-Abelian bridge, motivated by equivariant work in Attention, Fuchs et al. (2020); Liao & Smidt (2023); Xu et al. (2023), and Diffusion maps, Singer & Wu (2012). Unfortunately, the twisted-Hadamard-covariance for non-Abelian groups do not hold. For non-Abelian groups (e.g. SU(2)) acting by $E_{ij} \mapsto R_i E_{ij} R_j^{-1}$ with noncommuting $R_i$, the Hadamard map is not an equivariant bilinear map in general (componentwise, duplicated free indices and noncommuting middle factors obstruct rewriting the transform as $R_i(\cdot)R_j^{-1}$). This is why Abelian "twisted-Hadamard" constructions cleanly preserve equivariance, while their non-Abelian analogues require either promotion to tensor-squared connections (on $V \otimes V$) or projection to scalar invariants; see Appendix C.7.1.

### C.7.1 Nonabelian Equivariance

Let $G$ be a non-Abelian matrix group[1] with node-wise action on edge tensors

$$B_{ij,\alpha\beta} \longmapsto B'_{ij,\alpha\beta} = (R_i)_{\alpha\gamma} \, B_{ij,\gamma\delta} \, (R_j^{-1})_{\delta\beta}, \qquad R_i \in G,$$

and define the Hadamard (entrywise) product $(B \odot C)_{ij,\alpha\beta} := B_{ij,\alpha\beta} C_{ij,\alpha\beta}$. Then, in general,

$$(B \odot C)' \neq R_i \, (B \odot C) \, R_j^{-1},$$

i.e. the Hadamard map is not $G$–equivariant under left–right conjugation, unless the action reduces to one-dimensional characters (the Abelian case).

With $C$ transforming analogously to $B$, a direct calculation gives

$$(B \odot C)'_{ij,\alpha\beta} = (R_i)_{\alpha\gamma} \, (R_i)_{\alpha\mu} \, B_{ij,\gamma\delta} C_{ij,\mu\nu} \, (R_j^{-1})_{\delta\beta} (R_j^{-1})_{\nu\beta} . \tag{84}$$

Equivariance would require

$$(B \odot C)'_{ij,\alpha\beta} \overset{?}{=} (R_i)_{\alpha\rho} \, (B \odot C)_{ij,\rho\sigma} \, (R_j^{-1})_{\sigma\beta} = (R_i)_{\alpha\rho} \, B_{ij,\rho\sigma} C_{ij,\rho\sigma} \, (R_j^{-1})_{\sigma\beta}.$$

Comparing with equation 84 reveals two structural obstructions:

(i) Duplicated free indices. The factors $(R_i)_{\alpha\gamma}(R_i)_{\alpha\mu}$ (and similarly on the right) entail two independent copies of the representation at the same free index $\alpha$. To match the target form, one would need a $G$–equivariant bilinear map $V \otimes V \to V$ that canonically collapses $\gamma, \mu$ into a single index $\rho$ (and similarly $\delta, \nu$ into $\sigma$).

(ii) Noncommuting middle factors. Even abstractly attempting to factor equation 84 as $R_i(\cdot)R_j^{-1}$ fails because the two $R_i$'s (and two $R_j^{-1}$'s) act on different tensor legs and does not commute through $B$ and $C$ in a way that produces a single conjugation of $B \odot C$.

If $G = U(1)$ acts by characters $R_i = e^{i\phi_i}$, then entries pick up commuting phases and

$$(B \odot C)'_{ij} = R_i^{\alpha+\beta} \, (B \odot C)_{ij} \, R_j^{-(\alpha+\beta)},$$

so weights add and equivariance holds.

---

[1]With representation matrix $(R_i)_{\alpha\beta} = \exp\!\left(i\,\theta_a^i\,T_{\alpha\beta}^a\right)$ (matrix exponential), Lie algebra generators $T^a$, and node-dependent coefficients $\theta_a^i$.

# D EXPERIMENT DETAILS

## D.1 CIRCLE → TREFOIL POE SCHRÖDINGER BRIDGE EXPERIMENT

We describe the setup underlying Fig. 1. The goal is to realize a nonstationary Schrödinger bridge (NE regime) between two manifolds $L$ and $R$ sharing a latent parameter $t$, and to compare diffusion-based (DMAP) and attention-based (AMAP) PoE kernels under low-rank compression.

**Geometry and latent parameterization.** We fix $N = 400$ latent samples

$$t_i = \frac{i}{N}, \qquad i = 0, \dots, N-1,$$

and define two curves in $\mathbb{R}^3$:

$$L(t) = \big( \cos(2\pi t),\ \sin(2\pi t),\ 0 \big),$$
$$R(t) = \big( (2 + \cos(3 \cdot 2\pi t)) \cos(2 \cdot 2\pi t),\ (2 + \cos(3 \cdot 2\pi t)) \sin(2 \cdot 2\pi t),\ \sin(3 \cdot 2\pi t) \big),$$

corresponding to a unit circle and a trefoil knot, respectively. The point clouds

$$L_i := L(t_i), \qquad R_i := R(t_i)$$

are standardized coordinatewise to zero mean and unit variance for numerical stability.

**Endpoint marginals (NE bridge).** We construct nonstationary endpoint marginals $\mu^{\pm} \in \Delta^{N-1}$ as Gaussian bumps in the latent coordinate:

$$\tilde{\mu}_i^{\pm} := \exp\left( -\tfrac{1}{2} \left( \frac{t_i - c_{\pm}}{w} \right)^2 \right), \qquad \mu_i^{\pm} := \frac{\tilde{\mu}_i^{\pm}}{\sum_{k=1}^{N} \tilde{\mu}_k^{\pm}},$$

with centers $c_+ = 0.15$, $c_- = 0.65$ and common width $w = 0.07$. Since $\mu^+ \neq \mu^-$, the resulting SB is genuinely nonstationary (NE) in the sense of §4.0.1.

**DMAP PoE kernel.** We first build diffusion-maps kernels on $L$ and $R$. Let $D_L^2, D_R^2 \in \mathbb{R}^{N \times N}$ denote the squared Euclidean distance matrices

$$(D_L^2)_{ij} = \|L_i - L_j\|^2, \qquad (D_R^2)_{ij} = \|R_i - R_j\|^2.$$

For a fixed inverse temperature $\beta > 0$ (we use $\beta = 0.5$), we define row-stochastic diffusion kernels

$$(P_L)_{ij} = \mathrm{softmax}_j^+ \big( -\beta (D_L^2)_{ij} \big) = \frac{\exp(-\beta (D_L^2)_{ij})}{\sum_k \exp(-\beta (D_L^2)_{ik})},$$
$$(P_R)_{ij} = \mathrm{softmax}_j^+ \big( -\beta (D_R^2)_{ij} \big) = \frac{\exp(-\beta (D_R^2)_{ij})}{\sum_k \exp(-\beta (D_R^2)_{ik})}.$$

We then form the diffusion-based PoE kernel

$$K_{\mathrm{dmap}} := P_L \odot P_R, \qquad (K_{\mathrm{dmap}})_{ij} = (P_L)_{ij} (P_R)_{ij},$$

which is strictly positive and thus admissible as a Schrödinger reference kernel.

**AMAP PoE kernel.** For the AMAP-style kernel we use a Gram-based "to" divergence. Given $X \in \mathbb{R}^{N \times d}$, define

$$G := XX^\top, \qquad g := \mathrm{diag}(G), \qquad d_{ij}^{\rightarrow}(X) := g_j - G_{ij}.$$

We set $X = L$ and $X = R$ to obtain $d_L^{\rightarrow}, d_R^{\rightarrow} \in \mathbb{R}^{N \times N}$, and define row-stochastic attention kernels

$$(A_L)_{ij} = \mathrm{softmax}_j^+ \big( -\beta d_{ij}^{\rightarrow}(L) \big),$$
$$(A_R)_{ij} = \mathrm{softmax}_j^+ \big( -\beta d_{ij}^{\rightarrow}(R) \big),$$

with the same $\beta$ as above. The AMAP PoE kernel is then

$$K_{\mathrm{amap}} := A_L \odot A_R, \qquad (K_{\mathrm{amap}})_{ij} = (A_L)_{ij} (A_R)_{ij}.$$

**Full Schrödinger bridges and barycentric map.** For $K \in \{K_{\mathrm{dmap}}, K_{\mathrm{amap}}\}$ we solve the discrete SB problem

$$\Pi^\star = \arg \min_{\Pi \geq 0} \sum_{i,j} \Pi_{ij} \big( \log(\Pi_{ij}/K_{ij}) - 1 \big) \quad \text{s.t.} \quad \Pi \mathbf{1} = \mu^+, \ \Pi^\top \mathbf{1} = \mu^-,$$

via classical Sinkhorn iterations on the scaling vectors $u^+, u^- \in \mathbb{R}^N_{>0}$:

$$u^+ \leftarrow \mu^+ \oslash (Ku^-),$$
$$u^- \leftarrow \mu^- \oslash (K^\top u^+),$$

where $\oslash$ denotes componentwise division. After convergence we form

$$\Pi^\star = \mathrm{diag}(u^+) K \mathrm{diag}(u^-),$$

and verify that the row and column marginals match $\mu^\pm$ up to machine precision. The induced barycentric latent map $t \mapsto t'$ is computed as

$$t'_i := \frac{\sum_j \Pi^\star_{ij} t_j}{\sum_j \Pi^\star_{ij}}, \qquad i = 1, \ldots, N.$$

**Low-rank PoE SB via SVD factors.** To study compressibility, we approximate each directional expert $G_0, G_1 \in \mathbb{R}^{N \times N}$ (either $P_L, P_R$ or $A_L, A_R$) by a rank-$r$ truncated SVD. For a given $G$ we write

$$G = U \Sigma V^\top, \qquad \Sigma = \mathrm{diag}(s_1, \ldots, s_N), \quad s_1 \geq \cdots \geq s_N \geq 0,$$

and set

$$U_r := U_{:,1:r}, \quad \Sigma_r := \mathrm{diag}(s_1, \ldots, s_r), \quad V_r := V_{:,1:r},$$

together with

$$L := U_r \Sigma_r^{1/2}, \qquad R := \Sigma_r^{1/2} V_r^\top,$$

so that $G_{\mathrm{lr}} = LR$ is the best rank-$r$ approximation of $G$ in Frobenius norm. The PoE kernel is then approximated by

$$K_{\mathrm{lr}} := G_{0,\mathrm{lr}} \odot G_{1,\mathrm{lr}}.$$

We implement a Sinkhorn solver that accesses $K_{\mathrm{lr}}$ only through matrix–vector products with $z \in \mathbb{R}^N$:

$$K_{\mathrm{lr}} z \approx (G_{0,\mathrm{lr}} \odot G_{1,\mathrm{lr}}) z,$$
$$K_{\mathrm{lr}}^\top z \approx (G_{0,\mathrm{lr}} \odot G_{1,\mathrm{lr}})^\top z,$$

computed using the low-rank factors $L_0, R_0, L_1, R_1$ without ever forming $K_{\mathrm{lr}}$ explicitly (cf. the `poe_matvec` and `poe_matvec_T` routines in the code).

After convergence of low-rank Sinkhorn we reconstruct

$$\Pi_{\mathrm{lr}} = \mathrm{diag}(u^+_{\mathrm{lr}}) \, K_{\mathrm{lr}} \, \mathrm{diag}(u^-_{\mathrm{lr}})$$

for evaluation and compute the corresponding barycentric map $t \mapsto \hat{t}'$.

**Evaluation metrics.** For each method (DMAP vs. AMAP) and rank $r$ we report:

- *marginal errors*

$$\text{row\_err} := \max_i \big| \sum_j \Pi_{\mathrm{lr},ij} - \mu^+_i \big|, \qquad \text{col\_err} := \max_j \big| \sum_i \Pi_{\mathrm{lr},ij} - \mu^-_j \big|;$$

- *kernel relative error*

$$\frac{\|K_{\mathrm{lr}} - K\|_F}{\|K\|_F};$$

- *coupling relative error*

$$\frac{\|\Pi_{\mathrm{lr}} - \Pi^\star\|_F}{\|\Pi^\star\|_F};$$

- *barycentric latent RMSE*

$$\text{RMSE}(t') := \sqrt{\frac{1}{N} \sum_{i=1}^{N} (t'_i - \hat{t}'_i)^2},$$

where $t'$ is computed from the full SB and $\hat{t}'$ from the low-rank SB.

In the regime $r \in \{12, 13, 14\}$ we observe that AMAP PoE kernels admit substantially smaller kernel and coupling errors, as well as barycentric RMSE below $10^{-3}$ for $r \approx 12$, while DMAP PoE kernels require slightly higher ranks to achieve comparable accuracy. These numerical results support the claim that attention-style PoE kernels are highly compressible while preserving the induced Schrödinger-bridge transport in latent space.

### D.2 MINIMAL DDPMs WITH MARKOV DENOISERS ON A SWISS ROLL

We describe the setup underlying Fig. 2, where we compare AMAP- and DMAP-based Markov denoisers inside a DDPM on a 3D Swiss roll.

**Data.** We generate $N = 3000$ samples from the standard 3D Swiss roll using `make_swiss_roll` with noise level 0.1, obtaining points $x_i \in \mathbb{R}^3$. The resulting cloud is standardized coordinatewise to zero mean and unit variance:

$$\tilde{x}_i := \frac{x_i - \mu}{\sigma}, \qquad \mu := \frac{1}{N} \sum_{i=1}^{N} x_i, \quad \sigma_k := \sqrt{\frac{1}{N} \sum_{i=1}^{N} (x_{ik} - \mu_k)^2 + 10^{-6}}.$$

All training and evaluation are performed on the standardized data $\tilde{x}_i$.

**Diffusion schedule.** We use a cosine-based variance schedule following Nichol–Dhariwal: define

$$\bar{\alpha}_t^{\cos} \propto \cos^2\left( \frac{(t/T + s)}{1 + s} \frac{\pi}{2} \right), \qquad t = 0, \ldots, T,$$

with $T = 200$ and offset $s = 0.008$, normalized so that $\bar{\alpha}_0^{\cos} = 1$. We set

$$\bar{\alpha}_t := \bar{\alpha}_t^{\cos}, \quad \alpha_t := \frac{\bar{\alpha}_t}{\bar{\alpha}_{t-1}}, \quad \beta_t := 1 - \alpha_t,$$

and clip $\beta_t \leq \beta_{\max} = 0.02$, recomputing

$$\alpha_t := 1 - \beta_t, \qquad \bar{\alpha}_t := \prod_{s=1}^{t} \alpha_s.$$

Forward diffusion is given by

$$x_t = \sqrt{\bar{\alpha}_t}\, x_0 + \sqrt{1 - \bar{\alpha}_t}\, \varepsilon, \qquad \varepsilon \sim \mathcal{N}(0, I_3).$$

**Time-dependent temperature.** Both denoisers use a scalar inverse temperature $\lambda(t)$ that increases with the timestep index. For $t \in \{0, \ldots, T - 1\}$, we define

$$\lambda(t) = \lambda_{\min} + (\lambda_{\max} - \lambda_{\min}) \frac{t}{T - 1}, \qquad \lambda_{\min} = 0.3,\ \lambda_{\max} = 1.0.$$

In the implementation, the integer $t$ is broadcast to a batch vector $t_{\text{idx}}$ and mapped to $\lambda(t)$; this scalar then scales the logits of the Markov kernel at timestep $t$.

**AMAP denoiser (AMAP-DDPM).** The AMAP denoiser $\varepsilon_\theta^{\text{AMAP}}$ is built from attention-like AMAP blocks, following the NESS geometry of §4.2. Each input $\tilde{x} \in \mathbb{R}^3$ is viewed as a sequence of $D = 3$ "tokens" (coordinates). We first embed coordinates into a $d_{\text{model}}$-dimensional feature space:

$$h \in \mathbb{R}^{B \times D \times d_{\text{model}}}, \qquad h = \text{Dense}(d_{\text{model}})(\tilde{x}) + \text{pos\_emb},$$

where $\mathrm{pos\_emb} \in \mathbb{R}^{D \times d_{\mathrm{model}}}$ is a learned positional embedding. Each AMAP block applies layer normalization followed by a multi-head AMAP self-attention with residual connection:

$$h \leftarrow h + \mathrm{AMAPSelfAttn}(h, t),$$

and we stack $L = 2$ such blocks. Inside $\mathrm{AMAPSelfAttn}$ we compute standard Q,K,V projections:

$$Q = \mathrm{reshape\_heads}(\mathrm{Dense}(d_{\mathrm{model}})(h)),$$
$$K = \mathrm{reshape\_heads}(\mathrm{Dense}(d_{\mathrm{model}})(h)),$$
$$V = \mathrm{reshape\_heads}(\mathrm{Dense}(d_{\mathrm{model}})(h)),$$

with $H = 8$ heads and per-head dimension $d_h = d_{\mathrm{model}}/H$. The AMAP logits at time $t$ are

$$S_{ij}^{\mathrm{AMAP}}(t) = \lambda(t) \frac{\langle Q_i, K_j \rangle}{\sqrt{d_h}},$$

and the corresponding row-stochastic Markov kernel is

$$P_{ij}^{\mathrm{AMAP}}(t) = \frac{\exp(S_{ij}^{\mathrm{AMAP}}(t))}{\sum_m \exp(S_{im}^{\mathrm{AMAP}}(t))}.$$

The block output is given by

$$\tilde{h}_i = \sum_j P_{ij}^{\mathrm{AMAP}}(t) V_j,$$

followed by a head-merge and output projection back to $d_{\mathrm{model}}$. A final linear layer maps token features to a single scalar per coordinate, yielding the noise prediction $\varepsilon_\theta^{\mathrm{AMAP}}(x_t, t) \in \mathbb{R}^3$.

**DMAP denoiser (DMAP-DDPM).** The DMAP denoiser $\varepsilon_\theta^{\mathrm{DMAP}}$ shares the same architecture as above (same $d_{\mathrm{model}}$, $H$, $L$, embeddings, and MLP structure), but replaces the attention kernel by a QK-metric DMAP kernel, in line with the diffusion-map viewpoint of §4.1. Given Q,K as above, we form per-head squared norms

$$\|Q_i\|^2, \quad \|K_j\|^2, \qquad q_i := \|Q_i\|^2, \quad k_j := \|K_j\|^2,$$

and cross terms $C_{ij} := \langle Q_i, K_j \rangle$. The QK-based squared "distance" is defined symmetrically as

$$D_{ij}^2 = q_i + k_j - C_{ij} - C_{ji},$$

which satisfies $D_{ij}^2 = D_{ji}^2$ by construction. The DMAP logits and Markov kernel at time $t$ are

$$S_{ij}^{\mathrm{DMAP}}(t) = -\lambda(t) D_{ij}^2, \qquad P_{ij}^{\mathrm{DMAP}}(t) = \frac{\exp(S_{ij}^{\mathrm{DMAP}}(t))}{\sum_m \exp(S_{im}^{\mathrm{DMAP}}(t))},$$

so that each block implements a diffusion-map-like Markov operator over feature coordinates, built from a QK metric rather than direct similarity. As in the AMAP case, we set

$$\tilde{h}_i = \sum_j P_{ij}^{\mathrm{DMAP}}(t) V_j$$

per head, followed by head merge, projection, and a linear readout to obtain $\varepsilon_\theta^{\mathrm{DMAP}}(x_t, t) \in \mathbb{R}^3$.

**Training.** Both models are trained with the standard $\epsilon$-prediction loss

$$\mathcal{L}(\theta) = \mathbb{E}_{t, x_0, \varepsilon} \|\varepsilon_\theta(x_t, t) - \varepsilon\|^2,$$

where $t$ is sampled uniformly from $\{0, \dots, T-1\}$ and $x_t$ is generated as above. We use Adam with learning rate $10^{-3}$, batch size 256, and exponential moving average (EMA) of parameters with decay 0.999. The AMAP-DDPM is trained for $3 \times 10^4$ gradient steps. The QK-DMAP-DDPM converges more slowly and is therefore trained for $1.8 \times 10^5$ steps to achieve comparable visual quality and kernel MMD scores. In all cases, we use the EMA parameters for sampling.

**Sampling.** Reverse diffusion follows the standard DDPM update. Starting from $x_T \sim \mathcal{N}(0, I_3)$, we iterate for $t = T - 1, \ldots, 0$:

$$x_t = \mu_t(x_{t+1}, \varepsilon_\theta(x_{t+1}, t+1)) + \sigma_t z, \qquad z \sim \mathcal{N}(0, I_3),$$

where $\mu_t$ is the usual DDPM mean expressed in terms of $\alpha_t$, $\bar{\alpha}_t$, and the posterior variance

$$\tilde{\beta}_t = \frac{1 - \bar{\alpha}_{t-1}}{1 - \bar{\alpha}_t} \beta_t.$$

We generate $N_{\mathrm{sample}} = 5000$ samples from each model for visualization and evaluation.

**Evaluation metric.** To quantify sample quality we report an RBF maximum mean discrepancy ($\mathrm{MMD}^2$) between generated samples and the standardized training data. Given two sets $X = \{x_i\}_{i=1}^n$ and $Y = \{y_j\}_{j=1}^m$ in $\mathbb{R}^3$, we use the unbiased estimator

$$\mathrm{MMD}^2(X, Y) = \frac{1}{n(n-1)} \sum_{i \neq i'} k(x_i, x_{i'}) + \frac{1}{m(m-1)} \sum_{j \neq j'} k(y_j, y_{j'})$$

$$- \frac{2}{nm} \sum_{i,j} k(x_i, y_j),$$

with $k(x, y) = \exp\left(-\|x - y\|^2 / (2\sigma^2)\right)$ and $\sigma = 1$. The values reported in Fig. 2 ($\mathrm{MMD}^2$ in the titles of the sample panels) show that AMAP-DDPM attains comparable or slightly better $\mathrm{MMD}^2$ with an order of magnitude fewer optimization steps than the QK-DMAP-DDPM, consistent with the view that AMAP's NESS Markov geometry is a more efficient inductive bias for denoising than the purely diffusion-based QK-DMAP operator.

CONTENTS

## LIST OF THEOREMS, LEMMAS, COROLLARY AND DEFINITIONS

