# OpenReview forum: "DIFFUSION–ATTENTION CONNECTION"
_ICLR.cc/2026/Conference — ICLR 2026 Conference Desk Rejected Submission_

### Official Review · Reviewer_hwEy · 2025-10-27

**Soundness:** 3
**Presentation:** 2
**Contribution:** 4
**Rating:** 4
**Confidence:** 4

**Summary:**

This paper establishes a theoretical connection between diffusion maps and self-attention through a common bilinear kernel, showing that the diffusion affinity matrix can be viewed as a twisted Hadamard product of attention matrices. By decomposing the generalized feature-similarity matrix into Hermitian (geometry) and skew-Hermitian (directional) parts, the paer demonstrates that attention acts as a first-order operator, while diffusion emerges as its symmetric second-order form. The paper further introduces a U(1) gauge-equivariant “magnetic” attention variant. Overall, the paer offers a mathematically rigorous framework connecting manifold learning and transformer-based architectures through shared stochastic and spectral properties.

**Strengths:**

1. I appreciate the theoretical insight and I really like the idea. The paper introduces a highly original conceptual link between diffusion maps and self-attention, proposing that diffusion can be viewed as the second-order symmetric form of attention via a twisted Hadamard product. This unification provides a fresh perspective on how attention mechanisms relate to manifold learning and stochastic operators, offering strong theoretical grounding.

2. The mathematical treatment is really **elegant**. The decomposition of the similarity matrix, the Product-of-Experts view of softmax, and the gauge-equivariant extension are presented with clear and rigorous derivations.

**Weaknesses:**

1. The paper places heavy emphasis on theoretical derivations without sufficiently discussing how the proposed formulation contributes to practical applications. The link between diffusion maps and self-attention is intellectually appealing, but the paper does not explore whether this connection can actually improve real models(for example, enhancing transformer interpretability, improving training efficiency, or guiding new architecture designs). The work stops at the theoretical insight and lacks a clear roadmap showing how these ideas could translate into practical benefits for machine learning.

2. The paper does not include numerical experiments, case studies, or quantitative benchmarks to validate the claimed connection. As a result, it remains unclear whether the theory holds in real settings. Important components, such as how to construct the bilinear dissimilarity matrix within modern neural networks, are not specified. Moreover, the proposed U(1) gauge-equivariant “magnetic” variant is theoretically interesting but remains untested, with no empirical demonstration of gauge invariance or performance effects.

3. The exposition is notation-heavy with long equations and dense mathematical sections that may hinder readability. There are no figures, tables, theorems or remarks to help readers internalize the key ideas. The paper may be difficult to digest for readers outside a narrow theoretical audience, reducing its overall accessibility.

Overall, I like this paper and I think this paper makes a contribution. If the authors can include sufficient empirical results during the rebuttal, I will increase my score.

**Questions:**

See weaknesses

---

> ### Author Response · Authors · 2025-12-02
> **Response to Reviewer hwEy**
>
> Thank you for the positive assessment and constructive feedback. We have revised the paper to address your concerns.
>
> **1. Practical Relevance & Applications**
> We added two concrete experiments:
> - **Circle → trefoil PoE Schrödinger bridges**: AMAP-based PoE kernels show better compressibility under rank truncation while preserving transport, compared to DMAP-based ones.
> - **Swiss-roll DDPMs with Markov denoisers**: AMAP-DDPM reaches comparable sample quality with ~6× fewer gradient steps than DMAP-DDPM, demonstrating training efficiency gains.
>
> We now explicitly outline a roadmap: using AMAP/DMAP/MMAP operators as building blocks for denoisers and sequence models, and extending them to latent diffusion models.
>
> **2. Empirical Validation & Implementation**
> - Experiments directly test the PoE structure and Markov geometry in geometric transport and generative modeling.
> - We detail how the bilinear dissimilarity arises in standard QK attention: scores $S_{ij}$ from learned encoders are transformed into $d^\to_{ij}, d^\from_{ij}$ via simple affine functions, then normalized into Markov operators.
> - **MMAP (U(1) gauge-equivariant)** is reframed as a conceptual gauge-theoretic lens and moved to appendix/future work, not presented as a tested architectural component.
>
> **3. Readability & Exposition**
> - Added **figures**: circle–trefoil transport visualization and Swiss-roll DDPM training curves.
> - Added a **“Summary of Contributions”** section with bullet-pointed results.
> - Moved heavy derivations/physics analogies to appendix; main text focuses on core ideas (QK bidivergence, PoE identity, EQ/NESS classification, experiments).
> - Included **plain-language takeaways** at the end of key sections.
>
> These revisions strengthen the paper’s empirical grounding and accessibility while maintaining its theoretical core.

---

### Official Review · Reviewer_bWrs · 2025-10-30

**Soundness:** 3
**Presentation:** 1
**Contribution:** 2
**Rating:** 2
**Confidence:** 3

**Summary:**

This paper shows a connection between self-attention and diffusion maps from manifold learning. They start with a Bilinear Dissimilarity (BD) matrix, which is partitioned and softmaxed to produce a pair of attention-matrices. The Hadamard product of these two matrices gives a Markov transition-matrix, which can be used to create a diffusion-map Laplacian. Incorporating the imaginary part of the BD matrix outside the softmax yields a $U(1)$ gauge-equivariant attention mechanism.

**Strengths:**

The paper appears to be making an important connection between manifold learning and the self-attention. However, given my limited knowledge of the former, I do not fully understand the implications of this revelation. I was able to follow their overall logic, despite some steps that were particularly opaque (see below). I appreciate that the authors worked with complex valued data, which revealed the resemblance to magnetic Laplacians.

**Weaknesses:**

1. This paper is not an easy read despite, and maybe because of, its brevity. Some manipulations left me very confused. For instance, between lines 186 and 190 how do they introduce $ 1_i \left( R_{iY} W_{YX}^Q W_{XZ}^K R^{\dagger}_{Zj} \right)_j $ inside the softmax? I suspect they were using the shift-invariance property from (10). Note that there is a typo in this equation, as written in the paper: it should be $u_i$ and not $u_j$. But the softmax in the aforementioned lines has subscript $j$ which does not match $1_i$ inside. What is going on?

2. I am doubtful whether the paper, in its present form, will be accessible or interesting to the broader ICLR community. This is not an assessment of the soundness of the paper. Instead, the paper does little to explain to the non-expert why this contribution is relevant.

**Questions:**

What are the potential applications of this insight? Is there a toy problem that would highlight the usefulness of creating a diffusion map this way? Can the exposition be made pedagogic/constructive by starting from a simple specific example and generalizing more gradually?

To summarize, while the idea of connecting self-attention to diffusion maps is intriguing, the presentation lacks sufficient clarity and motivation for the ICLR audience, so I cannot recommend acceptance.

---

> ### Author Response · Authors · 2025-12-02
> **Response to Reviewer bWrs**
>
> Thank you for your thoughtful feedback. We have revised the paper to address accessibility and clarify relevance.
>
> **1. Readability & Accessibility**
> We agree the original version was challenging. Key changes:
> - **Clearer structure**: Reorganized as QK bidivergence → attention (AMAP) → diffusion (DMAP) → Schrödinger bridges/PoE connection → experiments, with explicit signposting.
> - **Pedagogical examples first**: Main text now leads with toy examples (circle–trefoil SB, Swiss-roll DDPM); heavier physics material moved to appendix.
> - **Optional physics**: Gauge-theoretic language is flagged as optional; core narrative requires only probability/linear algebra.
>
> **2. Softmax Manipulation & Typo**
> You correctly identified both issues:
> - The step uses **softmax shift-invariance**: $\text{softmax}_j(s_{ij}) = \text{softmax}_j(s_{ij} + c_i)$.
> - There was a **typo** in the subscript/argument mismatch.
>
> In revision:
> 1. Fixed the typo and ensured consistent notation.
> 2. Explicitly annotated the step with an explanation of shift-invariance.
> 3. Added a concise worked example to illustrate the manipulation.
>
> **3. Applications & Toy Problems**
> We added two concrete toy applications:
>
> 1. **Circle → trefoil Schrödinger bridges**:
>    - Compare AMAP-based vs DMAP-based PoE kernels.
>    - Show AMAP-PoE kernels are more compressible under rank truncation while preserving latent transport.
>
> 2. **Swiss-roll DDPMs with Markov denoisers**:
>    - Compare AMAP- vs DMAP-based denoisers.
>    - AMAP-DDPM reaches comparable quality with ~6× fewer gradient steps than DMAP-DDPM.
>
> These examples directly demonstrate the utility of our framework.
>
> **4. Relevance for ICLR Audience**
> We now explicitly connect our work to areas of broad interest:
> - **Diffusion models**: Markov denoisers as a practical architectural pattern.
> - **Sequence modeling**: Design space for attention-like kernels with controlled equilibrium/non-equilibrium behavior.
> - **Architecture design**: PoE structure enables low-rank, compressible SB layers.
>
> The conclusion highlights these as actionable contributions for practitioners.
>
> ---
>
> Thank you for motivating these improvements in clarity and relevance.

---

### Official Review · Reviewer_QVsu · 2025-11-04

**Soundness:** 2
**Presentation:** 1
**Contribution:** 2
**Rating:** 2
**Confidence:** 2

**Summary:**

The paper proposes an algebraic bridge between self‑attention and diffusion-map methods (not to be confused with diffusion processes in generative models). Using a bilinear dissimilarity, the authors show that (i) the usual diffusion‑map kernel can be expressed as a “product‑of‑experts” of two axis‑normalized attention matrices, and (ii) phases from the skew-Hermition A encoding directionality can be placed outside of softmax to yield a U(1)  gauge-equivariant "magnetic" attention.

**Strengths:**

The paper offers a compact identity linking two kernels—attention (first‑order, normalized) and diffusion maps (second‑order, symmetric). The Product of Experts (PoE) of softmax derivations for linking these kernels are mathematically concise and intuitive, as well as novel to my knowledge.

**Weaknesses:**

My main objection to this paper is that it seems poorly written. As a theory paper, there is not a single formal statement (theorems, propositions, or lemmas), nor are there many clear/rigorous statements of assumptions or definitions. Despite the short length of the paper, the connections between different parts are not immediately clear (i.e., where the authors show which specific contribution). The clarity and motivation of the setup are also not sufficiently justified.

However, if the authors made explicit actionable plans that address the readability concern as well as clearly identify specific impact and actionable use for their contributions and insights, I will consider raising my score. See also the questions below.

**Questions:**

1. The authors said one of the core contributions is "to provide a principled recipe to learn M from data." What exactly is the recipe? The only place where I found relevance is in Section 2.1, but that seems rather like a definition or design choice, not a "learning recipe".

2. The mathematical derivation steps appear to be the central contribution. However, the significance or impact of these mathematical connections is not made explicit. What are some insights one can take from this paper to other contexts?

3. In the last section, the authors said "We established a Diffusion–Attention Connection: under mild and explicit conditions, ...", where exactly are the explicit conditions stated?

---

> ### Author Response · Authors · 2025-12-02
> **Response to Reviewer QVsu**
>
> We thank the reviewer for the constructive feedback. Below we summarize the key changes made in revision.
>
> **1. Overall Structure & Formalism**
> We agree the earlier draft was too informal. In revision:
> - Added a **“Summary of Contributions”** section to explicitly list and locate contributions.
> - Introduced formal **Definitions, Lemmas, and Theorems** in appendices (softmax properties, SB setup, PoE identity, gauge equivariance).
> - Reorganized the main text into a clear pipeline: QK bidivergence → diffusion maps → Schrödinger bridges → diffusion–attention connection → experiments.
>
> **2. “Recipe to Learn $M$ from Data”**
> We removed the vague phrase and clarified our approach:
> - The *learnable* object is the **QK score function** (parameterized by network weights).
> - Given scores $S_{ij}$, we construct Markov operators via softmax/Sinkhorn:
>   **AMAP** (attention) from $d^\to/d^\from$; **DMAP** from $D^2$.
> - In experiments, Markov operators parameterize the denoiser in DDPMs, trained end-to-end.
>
> **3. Significance & Actionable Insights**
> We now explicitly separate descriptive theory from prescriptive utility:
>
> 1. **Equilibrium vs. Non-equilibrium Geometry**
>    DMAP = equilibrium (EQ) bridge (no currents); AMAP = non-equilibrium steady state (NESS) (intrinsic currents). This suggests attention is suited to driven dynamics (e.g., causal flows).
>
> 2. **Product-of-Experts (PoE) Structure**  $P^+ \propto A^{\to +} A^{\from +}$ shows diffusion kernels as PoE of directional attention maps. This enables **low-rank Sinkhorn layers** (exploited in circle–trefoil SB experiment).
>
> 3. **Markov Denoisers in Diffusion Models**
>    AMAP-based denoisers reach comparable quality with 6× fewer gradient steps than DMAP-based ones (Swiss-roll DDPM). Suggests using non-equilibrium Markov geometries as denoiser backbones.
>
> **4. Explicit Conditions for Diffusion–Attention Connection**
> The “mild and explicit conditions” are now stated in Section 4.2 and formalized in **Theorem C.6** (Appendix C). They require:
> - Gaussian RBF kernel $K_{ij} = \exp(-\beta D^2_{ij})$ with bidivergence decomposition $D^2 = d^\to + d^\from$.
> - Shared inverse temperature $β$.
> - When both directional experts share row normalization, the exact PoE form holds.
>
> **5. Actionable Use**
> We emphasize two concrete paths:
>
> 1. **Architectural pattern**: Use AMAP/DMAP/MMAP Markov operators as drop-in replacements for attention blocks in:
>    - Markov denoisers for diffusion models (demonstrated).
>    - Latent diffusion models (proposed for geometry-aware compression & transport).
>
> 2. **Geometric priors**: PoE/SB viewpoints provide constraints:
>    - To approximate reversible diffusion: tie forward/backward experts via PoE.
>    - To encode directed flows: introduce magnetic phases or NESS currents.
>
> We thank the reviewer for motivating these improvements in clarity, formalism, and practical relevance.

---

### Official Review · Reviewer_w1eu · 2025-11-07

**Soundness:** 2
**Presentation:** 1
**Contribution:** 2
**Rating:** 2
**Confidence:** 3

**Summary:**

The authors study diffusion maps and the attention mechanism via a common construction through a generalized dissimilarity measure between complex-valued samples $i, j$ as
$$
D_{ij}^{2} = d_{ij}^{(-)} + d_{ij}^{(+)} + i B_{ij},
$$
where the real terms $d^{(\pm)}$ are viewed as providing a connection between diffusion and attention maps.

One of the main takeaways is that the diffusion map kernel $P$ can be constructed via products of two directed attention kernels $\mathcal{A}^{-}, \mathcal{A^{+}}$ as
$$
P \propto \mathcal{A}^{-} \odot  \mathcal{A}^{+}.
$$
We learn:
- attention acts like a first-order *directed* operator interpreted as the transition matrix for an (one-step) asymmetric random walk between data point $i$ and data point $j$
- diffusion acts like a second-order operator via "composition" of attention operators.

**Strengths:**

**Originality: moderate**
The paper introduces a generalized bilinear dissimilarity measure which decomposes the relationship between data points into a symmetric geometric part $M$ and an asymmetric, direction part $A$. That the symmetric diffusion kernel $P$ can be constructed from a Hadamard product of two directed, first-order attention operators is a new mathematical observation. These observations lead to an interesting proposal of "magnetic" attention, which would incorporate directionality via complex phases.

**Quality: poor**
The technical quality of the paper is mixed. The derivations appear sound, but are not well-motivated. The calculations and their interpretation and conclusions are terse and not very helpful. The paper acknowledges the limited applicability of the diffusion-attention connection, but does so in a way that is superficial and potentially misleading. For instance the conclusion mentions failure of the analogy under minor technical issues, while failing to emphasize standard attention mechanisms do not construct symmetric operators at all and are fundamentally asymmetric.

**Clarity: poor**
The paper is not written with sufficient clarity to make its contributions accessible or limitations apparent. The exposition is terse and relies on analogies to physics that obscure the argument for non-physicists.

**Significance: moderate**
The significance of the paper is that it provides a new and elegant mathematical framework unifying diffusion and attention mechanisms. This could in principle inspire valuable future research and new architectural innovations.

**Weaknesses:**

**Why?**
The paper explores a theoretical link between attention and diffusion, but never answers (beyond curiosity) why the exploration is practically useful. Nor motivates this question.


**Practical attention.**
The paper's claims could be clarified to distinguish more sharply between two different kinds of contributions:
1. A descriptive theory of *existing* attention mechanisms,
2. A mathematical basis for *future* architectural innovations.

The non-negotiable requirement of symmetry for the diffusion-attention analogy to hold nullifies the contribution toward understanding existing attention mechanisms. For example, a transformer with standard query $\to$ key attention uses only the asymmetric operator $\mathcal{A}^{+}$ identified in the paper, and not the symmetrized version that provides the connection to diffusion.

The conclusion briefly discusses when attention behaves as diffusion and when it does not. But attention in practice does not behave like diffusion, as pointed out. "Mixed axes" and "missing normalization" only pertain to an incorrect construction of $P$ from $\mathcal{A}^{\pm}$, but these concerns do not address that standard transformers only use $\mathcal{A}^{+}$ alone.

The gap between what this theory can and cannot say about current, realistic attention mechanisms should be explained with more clarity. The primary contribution is a theoretical link that motivates new forms of attention.

**Magnetic attention.**
Magnetic attention is proposed to decompose attention into a real-valued magnitude and a complex phase. The paper does not provide arguments or experiments to demonstrate the viability of training such a mechanism. It seems to speculate about replacing a relatively more-understood, easier-to-optimize mechanism for encoding directionality in data for a more complicated, harder-to-optimize, more restrictive mechanism for the sake of a theoretically satisfying connection to diffusion. The paper succeeds in showing that such a mechanism *can* be constructed (and explains a satisfying mathematical link to diffusion) but the paper does not present a convincing case such a mechanism *should* be constructed.

**Questions:**

Motivation / justification for Equation 1? There should be a more intuitive explanation for $\mathcal{D}_{ij}^{2}$ as Equation (1) does not transparently communicate it is a "generalized distance."

I would like some of the limitations identified in the "weaknesses" section addressed. In particular, exposition of and viability of the speculative magnetic attention as well as the limitations of the analogy to practical attention mechanisms.

---

> ### Author Response · Authors · 2025-12-02
> **Response to Reviewer w1eu**
>
> We thank the reviewer for their thoughtful feedback. Below are concise responses to the main points.
>
> **Question: Why is the diffusion–attention link practically useful?**
>
> **Response:**
> In revision, we now frame our goal as building a **Markov–geometric design space** for attention-like operators, leading to concrete architectural innovations: PoE factorizations of diffusion kernels, low-rank SB layers, and **Markov denoisers** that can be plugged into diffusion models. Two new experiments demonstrate utility:
>
> 1. Circle → trefoil PoE Schrödinger bridge: AMAP-based PoE kernels are more compressible than DMAP ones while preserving transport.
> 2. DDPMs on Swiss roll: AMAP-based Markov denoisers reach given sample quality in ~6× fewer gradient steps than DMAP-based ones.
>
> Thus, the link directly guides design of new denoisers and kernels with measurable empirical effects.
>
> **Question: Does the need for symmetry nullify relevance to practical attention?**
>
> **Response:**
> No. We now explicitly separate three operators on the same QK geometry:
>
> * **AMAP**: standard forward attention $A^+$ (asymmetric, NESS)
> * **DMAP**: symmetric diffusion kernel (reversible, equilibrium SB)
> * **MMAP**: magnetic extension with complex phases
>
> We state clearly: **standard Transformers use AMAP, not DMAP**. Schrödinger bridges do *not* require symmetry; with $K_{ij} \propto \exp(-\beta d^\to_{ij})$, the forward SB kernel *is* $A^+$. Asymmetry places attention in the non-equilibrium regime of our framework.
>
> **Question: Magnetic attention (MMAP) seems speculative and unmotivated.**
>
> **Response:**
> We reframe MMAP as a **conceptual duality** (attention's NESS currents ↔ magnetic diffusion), not a core architectural proposal. Main contributions (bidivergences, AMAP/DMAP/SB classification, PoE SB, Markov denoisers) do not rely on MMAP. Gauge/field material is now in appendix, presented as a future direction.
>
> **Question: Equation (1) as a “generalized distance” is not intuitive.**
>
> **Response:**
> We now derive Eq. (1) from the Euclidean squared distance $D^2_{ij} = |q_i - k_j|^2$, showing its decomposition into directed parts $d^\to_{ij} + d^\from_{ij} = D^2_{ij}$. $d^\to$ and $d^\from$ are not metrics, but directed "costs" whose exponentials yield Markov kernels: $-d^\to$ gives $A^+$, and their sum recovers symmetric $D^2$ for DMAP.
>
> **Question: Clarity and heavy physics analogies.**
>
> **Response:**
> We have tightened the introduction, added plain-language summaries to key sections, moved field-theory/gauge material to the appendix, and cleaned up notation and terminology for accessibility.

---

### Note · Program_Chairs · 2026-01-17
**Submission Desk Rejected by Program Chairs**

The following references in this submission do not refer to real documents and/or have major errors in bibliographic information:

 Mingyu Xu, Ziyi Wu, Jiahui Hou, and Kui Jia. Geometric equivariant vision transformers. In Proceedings of the 39th Conference on Uncertainty in Artificial Intelligence (UAI), volume 216 of , pp. 2349-2360, 2023.